# ATG8 delipidation is not universally critical for autophagy in plants

Yong Zou[1,9], Jonas A. Ohlsson [1,9], Sanjana Holla[1], Igor Sabljić [1], Jia Xuan Leong [2,3,4], Florentine Ballhaus [1], Melanie Krebs[3], Karin Schumacher[3], Panagiotis N. Moschou[1,5,6], Simon Stael[1], Suayib Üstün [7], Yasin Dagdas [8], Peter V. Bozhkov [1] & Elena A. Minina [1,3] ✉

Intracellular recycling via autophagy is governed by post-translational modifications of the autophagy-related (ATG) proteins. One notable example is ATG4-dependent delipidation of ATG8, a process that plays critical but distinct roles in autophagosome formation in yeast and mammals. Here, we aim to elucidate the specific contribution of this process to autophagosome formation in species representative of evolutionarily distant green plant lineages: unicellular green alga *Chlamydomonas reinhardtii*, with a relatively simple set of *ATG* genes, and a vascular plant *Arabidopsis thaliana*, harboring expanded *ATG* gene families. Remarkably, the more complex autophagy machinery of *Arabidopsis* renders ATG8 delipidation entirely dispensable for the maturation of autophagosomes, autophagic flux, and related stress tolerance; whereas autophagy in *Chlamydomonas* strictly depends on the ATG4-mediated delipidation of ATG8. Importantly, we also demonstrate the distinct impact of different Arabidopsis ATG8 orthologs on autophagosome formation, especially prevalent under nitrogen depletion, providing new insight into potential drivers behind the expansion of the ATG8 family in higher plants. Our findings underscore the evolutionary diversification of the molecular mechanism governing the maturation of autophagosomes in eukaryotic lineages and highlight how this conserved pathway is tailored to diverse organisms.

Autophagy is a catabolic pathway that plays a vital role in sustaining the functionality of cellular constituents and in helping organisms overcome nutrient scarcity. Autophagy balances out the biosynthetic activity of cells and salvages nutrients by sequestering expendable or dysfunctional cytoplasmic content into specialized double-membrane vesicles, autophagosomes, and delivering them to the lytic compartment for degradation and recycling[1]. Effective cargo sequestration relies on continuous expansion of the autophagosomal membrane to create adequately sized vesicles. Subsequently, these autophagosomes must be sealed before fusing with the lytic compartment to avoid the leakage of hydrolases into the cytoplasm[2].

The process of autophagosome formation and maturation is orchestrated by the autophagy-related proteins (ATGs) conserved among eukaryotes[1,3]. Post-translational modifications (PTMs) of the core ATGs regulate their assembly into protein complexes, which execute the sequential steps of autophagosome biogenesis in a timely

[1]Department of Molecular Sciences, Uppsala BioCenter, Swedish University of Agricultural Sciences and Linnean Center for Plant Biology, Uppsala, Sweden. [2]Department of Algal Development and Evolution, Max Planck Institute for Biology Tübingen, Tübingen, Germany. [3]Centre for Organismal Studies (COS), Heidelberg University, Heidelberg, Germany. [4]Center for Plant Molecular Biology (ZMBP), University of Tübingen, Tübingen, Germany. [5]Department of Biology, University of Crete, Heraklion, Greece. [6]Institute of Molecular Biology and Biotechnology, Foundation for Research and Technology-Hellas, Heraklion, Greece. [7]Faculty of Biology and Biotechnology, Ruhr-University Bochum, Bochum, Germany. [8]Gregor Mendel Institute of Molecular Plant Biology, Austrian Academy of Sciences, Vienna, Austria. [9]These authors contributed equally: Yong Zou, Jonas A. Ohlsson. ✉e-mail: alena.minina@slu.se

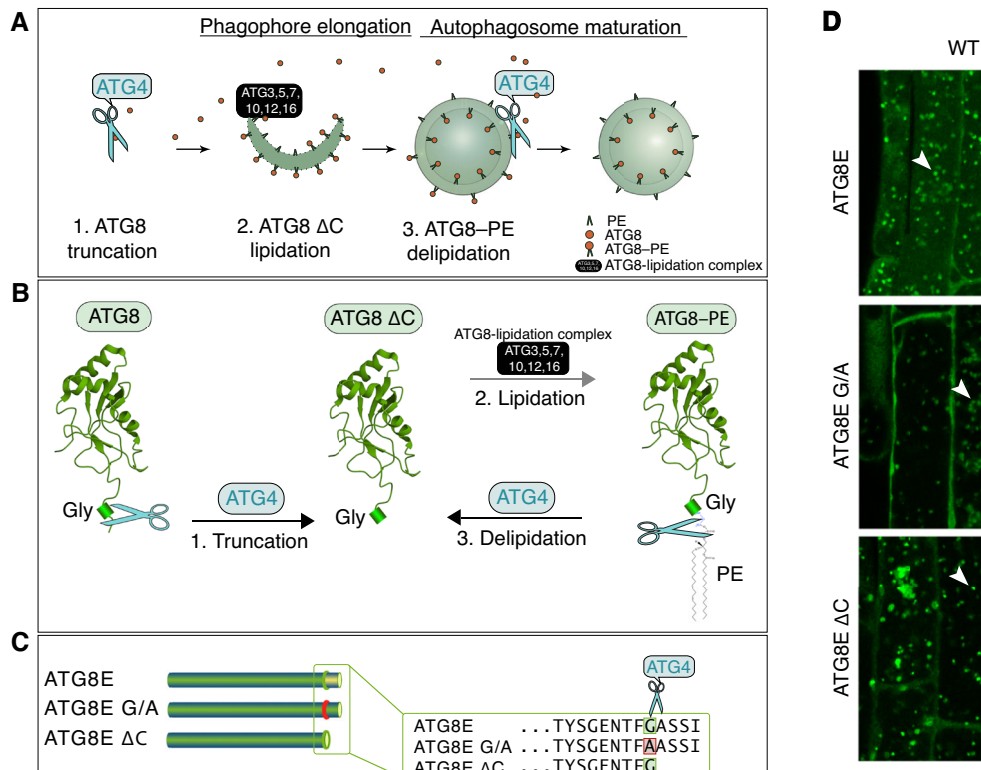

**Fig. 1 | Post-translational modifications of Arabidopsis ATG8 during autophagy. A** During autophagosome formation, ATG8 undergoes reversible conjugation with lipids present on the autophagosomal membranes. Lipidation of ATG8 onto phagophore membranes is essential for phagophore elongation. This process begins with the truncation of ATG8 through proteolytic processing by the ATG4 protease followed by lipidation aided by other core ATG proteins. Later, lipidated ATG8 is removed from the outer autophagosomal membrane during the maturation of the vesicle. **B** ATG8 is constitutively processed by ATG4 to remove the C-terminal peptide. The obtained truncated ATG8 (ATG8ΔC) has the C-terminal Gly residue exposed for the lipidation reaction. Lipidation of ATG8, i.e., its covalent conjugation to the phosphatidylethanolamine (PE) lipid present in the phagophore membranes (formation of ATG8–PE), is executed by ATG8-lipidation complex, best known as two ubiquitin-like conjugation systems, comprising several autophagy-related proteins, including ATG5 and ATG7. Delipidation of ATG8–PE on the outer membrane of autophagosome is performed by the ATG4 protease, which cleaves the amide bond between PE and ATG8, releasing the ATG8ΔC back into the pool of cytoplasmic ATG8 available for autophagosome biogenesis. **C** Diagram showing C-terminal sequences of the three artificially created variants of the Arabidopsis isoform ATG8E: ATG8E, full-length protein sequence including the C-terminal peptide masking the critical Gly residue; ATG8E G/A, full-length protein in which the Gly residue was replaced with Ala; ATG8EΔC, artificially truncated ATG8 protein lacking the C-terminal peptide that was masking the Gly residue. **D** Detection of three fluorescently labeled ATG8E variants (shown in **C**) in *Arabidopsis thaliana* epidermal root cells of 7-day-old seedlings treated with AZD8055 and concanamycin A. WT wild-type, *atg4a/b atg4a-2/b-2* double knockout of ATG4A and ATG4B genes. Arrowheads indicate autophagic bodies inside the vacuoles. The experiment was repeated twice. Scale bars, 20 μm.

manner. For example, the ubiquitin-like ATG8 protein undergoes a series of PTMs critical for the elongation of the nascent autophagosomal membrane, cargo sequestration, and recruitment of other core ATGs (Fig. 1A, B). The C-terminal peptide of ATG8 is removed by the dedicated protease ATG4, producing the most abundantly present adduct, ATG8 ΔC. Such truncated ATG8 (ATG8 ΔC) has its C-terminal glycine (Gly) residue exposed for subsequent modifications[4]. Upon upregulation of autophagy, the exposed Gly of ATG8 ΔC is conjugated with the lipid phosphatidylethanolamine (PE) present on the nascent autophagosomal membranes (Fig. 1B)[4,5]. The ATG8–PE conjugate decorates the inner and outer membranes of the forming autophagosome, playing crucial roles in cargo sequestration, as well as in the formation, closure, and fusion of autophagosomes[6]. At the later stages of vesicle biogenesis, ATG4 associates with the outer membrane of the autophagosome where it cleaves the amide bond between ATG8 and PE, thereby releasing back into the cytoplasmic pool the ATG8 ΔC together with other core ATGs recruited to the membrane by ATG8 (Fig. 1A, B). This process has two major functions: it facilitates the reuse of the released proteins for the formation of new autophagosomes, and it modifies the protein shell of the maturing autophagosome[7–9]. Delipidation of ATG8 has been reported to be critical for autophagy in yeast, animals, and plants[6,10,11]. Intriguingly, disruption of this

conserved step stalls different stages of autophagosome biogenesis in yeast and mammalian cells. Namely, ATG4-dependent ATG8 delipidation in animal cells is critical for autophagosomal docking and fusion with lysosomes[6,12]. However, abrogation of ATG8 delipidation in yeast impairs an earlier step of autophagosomal membrane elongation[10,13]. This discrepancy indicates fundamental differences in the maturation of yeast and animal autophagosomes and calls for further research necessary for mechanistic comprehension of the variation in the autophagosome biogenesis process among different eukaryotes.

We initiated this study with the aim of elucidating what step of autophagosome biogenesis in plants is affected by the lack of ATG8 delipidation. To our surprise, we discovered that autophagy of the vascular plant *Arabidopsis thaliana* can be carried out to its completion without ATG8 delipidation. Here we provide a thorough verification of the observed phenomenon by combining cell biology, biochemistry, and reverse genetics, and confirm the physiological relevance of the results by plant phenotyping. Additionally, we show that delipidation remains critical for autophagy of the evolutionarily distant to Arabidopsis unicellular green alga *Chlamydomonas reinhardtii*, which possesses simpler autophagic machinery compared to vascular plants. Lastly, we demonstrate that unlike artificially truncated Arabidopsis ATG8E

ΔC and ATG8F ΔC isoforms that can fully rescue autophagic flux in the absence of ATG4 activity, the natively truncated ATG8I isoform can rescue autophagic activity only partially. These results provide new insights into ATG8 isoform-specific roles in plant autophagosome formation.

## Results

### Expression of ATG8E ΔC restores accumulation of vacuolar puncta in ATG4-deficient Arabidopsis

To identify the specific stage in the process of plant autophagosome formation where ATG8 delipidation plays a crucial role, we produced ATG8-based autophagosomal fluorescent markers in the wild-type (WT) and ATG4-deficient (atg4a/b) backgrounds of the vascular plant model organism *A. thaliana*. The Arabidopsis genome codes for nine orthologs of ATG8, referred to as isoforms ATG8A–ATG8I[14]. Since the ATG8E isoform has been previously successfully used as a marker for autophagic activity in Arabidopsis[15,16], we implemented it for this study as well. We generated stable Arabidopsis lines expressing three variants of EosFP-tagged ATG8E: a full-length ATG8, a G/A lipidation-deficient mutant with the critical Gly residue substituted to Ala, and an artificially truncated ATG8 ΔC, lacking the C-terminal peptide that would normally mask the critical Gly residue from lipidation (Fig. 1B, C). When expressed in the atg4a/b background, ATG8 ΔC should readily form the ATG8–PE adduct anchored in the nascent autophagosomal membranes, however, the latter should not be delipidated in the absence of ATG4 activity.

The seedlings of the established marker lines were simultaneously treated with the autophagy-inducing compound AZD8055 (AZD), and with the compound concanamycin A (ConA) which increases vacuolar pH resulting in the deactivation of hydrolases and therefore preservation of the autophagosomes delivered to the plant vacuoles[17]. Expectedly, upon upregulation of autophagy in the WT plants, full-length ATG8E was incorporated into autophagosomes and delivered to the vacuoles, where it was detectable in the form of fluorescent puncta (Fig. 1D and Supplementary Fig. 1). The same marker remained dispersed throughout the cytoplasm in the atg4a/b background upon upregulation of autophagy, confirming the critical importance of ATG4-dependent processing for the ATG8's involvement in autophagosome formation. Lower intensity puncta were observed in the vacuoles of WT cells expressing the G/A mutant of ATG8 (Fig. 1D and Supplementary Fig. 1), indicating that it might be taken up from the cytoplasm as a cargo. Such puncta were not observed in the vacuoles of atg4a/b cells, corroborating the lack of autophagic activity in this genetic background. Surprisingly, the artificially truncated version, ATG8E ΔC, was observed in the vacuolar puncta of both WT and atg4a/b (Fig. 1D and Supplementary Fig. 1), suggesting that the absence of ATG8 delipidation in the latter case did not impede autophagic activity.

Given the unexpected nature of the observation described above, we found it necessary to confirm that the atg4/b mutant background used for this study is indeed ATG4-deficient. For this, we performed a thorough verification of the plants using genotyping, RT-PCR, and qPCR followed by an *in planta* ATG8-cleavage assay (Supplementary Fig. 2). The results of these assays confirmed the presence of the expected T-DNA insertions disrupting ATG4A and ATG4B genes (Supplementary Fig. 2A), leading to the loss of the corresponding transcripts (Supplementary Fig. 2C) and, subsequently, to the loss of ATG4 proteolytic activity (Supplementary Fig. 2E–G). qPCR analysis revealed a 7-fold increase in the expression of full-length ATG8E in atg4a/b (Supplementary Fig. 2D) yet no autophagic bodies were detectable in these plants (Fig. 1D). In contrast, a relatively lower expression of the artificial truncated ATG8E ΔC (Supplementary Fig. 2D) was sufficient for the formation of the autophagic bodies in the ATG4-deficient background (Fig. 1D).

### The formation of plant autophagosomes and their delivery to the vacuole do not depend on ATG8 delipidation

To verify that the EosFP-positive puncta observed in the vacuoles of atg4a/b plants were indeed autophagic bodies labeled with the EosFP–ATG8–PE adduct, we expressed EosFP–ATG8E ΔC in the background lacking ATG8 lipidation activity, i.e., knockout of ATG5 gene[18]. Transgenic plants expressing the artificially truncated version of ATG8E were subjected to AZD/ConA treatment prior to imaging the root and shoot epidermal cells. Upon treatment, the marker localized to vacuolar puncta in the WT and atg4a/b, but not in the atg5 plants, confirming that punctate localization of the marker was dependent on ATG8 lipidation and therefore represented autophagosomal structures (Fig. 2A). Quantification of puncta revealed that autophagic bodies accumulated in the vacuoles of atg4a/b cells in amounts comparable to WT, indicating reconstituted autophagic activity in the absence of the ATG8-delipidation step (Fig. 2B). Furthermore, we conducted a similar analysis on true leaves and confirmed that the observed phenomenon was not restricted to the early developmental stages of Arabidopsis but also occurred in adult plants (Supplementary Fig. 3A, B).

Next, we corroborated the results obtained via genetic ablation of ATG4 by using chemical inhibitors of ATG4 activity. For this, 10-day-old seedlings and one-month-old plants expressing EosFP–ATG8E ΔC in the WT background were treated with AZD and ConA supplemented with cysteine protease inhibitors iodoacetamide (IAM) or N-ethylmaleimide (NEM), chemical compounds previously shown to irreversibly inhibit ATG4 activity[19–21]. Neither IAM nor NEM treatment prevented the accumulation of autophagic bodies in the vacuoles of roots or shoots (Fig. 2C, Supplementary Fig. 3C, D, and Supplementary Movie 1), although some decrease in the number of puncta was observed in highly stressed cells. Due to the strong cytotoxicity of these chemical compounds, we decided to adhere to the use of genetic mutants for further *in planta* experiments. Additionally, we tested another inhibitor of cysteine proteases, E64d which was suggested to not impact ATG4 activity but potentially stabilize autophagic bodies in the lytic compartment[22]. Accordingly, we did not observe a significant difference between WT and atg4a/b mock- or E64d-treated plants expressing EosFP–ATG8E ΔC (Supplementary Fig. 3E, F). Western blot detection of EosFP–ATG8EΔC in the plants subjected to AZD/ConA treatment with or without E64d revealed accumulation of the ATG8–PE adduct in the ATG4-deficient background independently of treatment. However, treatment with E64d rendered ATG8–PE detectable in the WT protein extracts, most probably due to the better stability of the adduct under these conditions (Fig. 2D). The ATG8–PE band was not detectable in the protein extracts of atg5, consistent with the lack of ATG8-lipidating activity in this mutant (Fig. 2D). Finally, we validated that the lower band identified in Western blots indeed represents the ATG8–PE adduct by conducting an in vitro ATG8 delipidation assay (Supplementary Fig. 3G, H).

A study carried out in yeast revealed an APEAR motif (ATG8–PE association region) responsible for ATG4 recruitment to the lipidated form of ATG8 during autophagosome maturation[9]. Intriguingly, the APEAR motif is conserved in the ATG4B but not the ATG4A Arabidopsis ortholog[9], indicating the potential functional diversification of these orthologs in ATG8 processing. Importantly, individual knockouts of ATG4 in Arabidopsis do not display a discernible autophagy-deficient phenotype[8,23] and accumulate autophagic bodies in WT-like quantities upon AZD/ConA treatment (Supplementary Fig. 3I, J). If the two ATG4 enzymes indeed differ in their contribution to ATG8 delipidation, the lack of distinct phenotypes in the corresponding knockouts could provide further evidence for ATG8 delipidation not being critical for autophagy. To verify this hypothesis, we tested if delipidation efficacy is reduced in the single knockout for ATG4B bearing APEAR motif but not in the single ATG4A knockout when compared to WT. For this, we crossed atg4a/b and WT plants expressing EosFP–ATG8E ΔC and performed Western blot detection of the

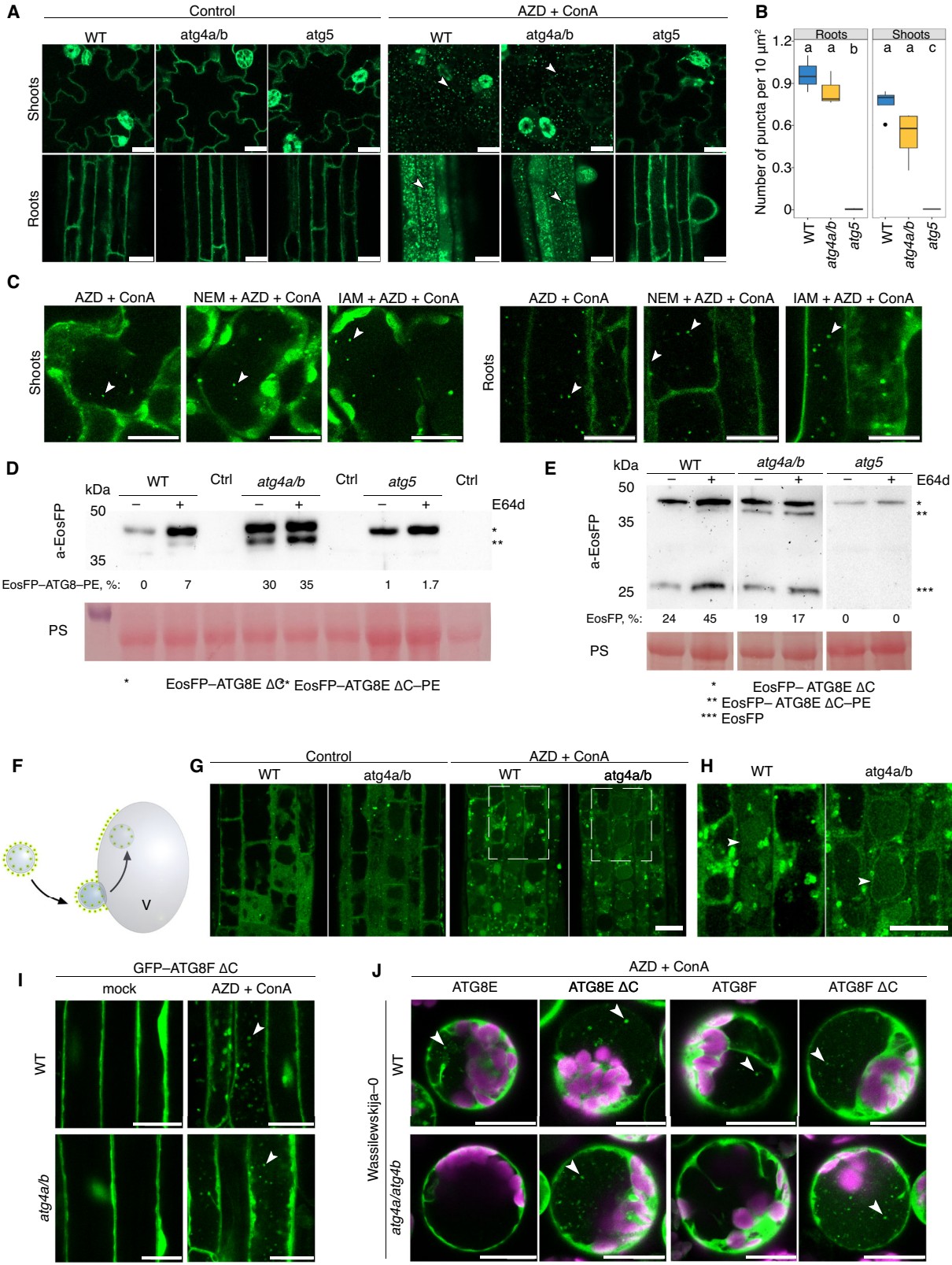

tagged ATG8 in the young (2-week-old) and older (6-week-old) plants. To our surprise, delipidation was significantly reduced in both individual *ATG4* knockouts (Supplementary Fig. 3K, L). Despite an at least 4-fold decrease in ATG8-delipidating activity in the single ATG4 knockouts compared to WT (Supplementary Fig. 3K, L), no difference in the number of autophagic bodies was observed in these backgrounds upon induction of autophagy (Supplementary Fig. 3I, J). These results are in agreement with the other observations pointing towards ATG8 delipidation being dispensable for plant autophagy. They also demonstrate that while the proteolytic activity required for C-terminal processing of ATG8 can be efficiently executed by a single ATG4 isoform (Supplementary Fig. 2E, F), the delipidating activity of individual Arabidopsis ATG4s is significantly less efficient and requires expression of both genes.

**Fig. 2 | Delipidation of ATG8E and F is dispensable for autophagosome formation and vacuolar delivery.** The effects of artificially truncated ATG8EΔC (**A–H**) and ATG8FΔC (**I**) isoforms in the *atg4a-2/b-2* knockout are also reproducible in the *atg4a-1/atg4b-1* knockout (**J**). **A** CLSM of Arabidopsis seedlings roots and shoots expressing EosFP-ATG8EΔC in WT, *ATG4*- or *ATG5*-knockouts upon AZD/ConA treatment. **B** Quantification of vacuolar puncta (*n* = 12 shoots and 9 roots) for samples illustrated in (**A**). Statistics description is available in the "Methods". **C** CLSM of Arabidopsis shoots (left) and roots (right) expressing EosFP–ATG8EΔC in WT upon AZD/ConA treatment with or without cysteine protease inhibitors (1 mM IAM or 10 mM NEM). Accompanying time-lapse data are in Supplementary Movie 1, vacuolar puncta quantification is in Supplementary Fig. 3C, and the inhibitory effect of NEM and IAM treatments on ATG4 activity is shown in Supplementary Fig. 3D. **D** E64d treatment elevates accumulation of EosFP–ATG8EΔC–PE. Ctrl, negative control without transgene. Densitometry results, shown as numbers under the Western blot lanes, represent the integrated density of the ATG8-PE band as a percentage of the total signal intensity for each sample. **E** EosFP-cleavage assay performed on the same material used for (**D**) shows no decrease in autophagic activity in the samples accumulating ATG8–PE. Numbers under Western blot lanes represent densitometry results corresponding to the integrated density of the free EosFP band expressed as % of the total signal intensity for the corresponding sample. **F** Schematic representation of the expected EosFP signal accumulation on the tonoplast upon its fusion with EosFP–ATG8EΔC–decorated autophagosomes in the absence of ATG8-delipidation activity. **G** CLSM of Arabidopsis roots expressing EosFP–ATG8EΔC shows accumulation of tonoplastic EosFP signal upon upregulation of autophagy in *atg4a/b*. **H** Zoomed-in images of selected areas in (**D**) with adjusted contrast. **I** CLSM of WT and *atg4a-2/b-2* Arabidopsis seedling roots expressing GFP-ATG8FΔC treated with AZD/ConA. Vacuolar puncta quantification in Supplementary Fig. 4B. **J**. CLSM of protoplasts from *atg4a-1/4b-1* plants expressing GFP-tagged full length and truncated versions of ATG8E and ATG8F, treated with AZD/ConA. The complete data set is in Fig. S7. White arrowheads indicate autophagic bodies (**A, C, I, J**) or tonoplast (**H**). Scale bars, 20 μm. PS, Ponceau S.

To corroborate the reconstitution of autophagic activity in the absence of ATG8 delipidation, we deployed a modified version of the GFP cleavage assay[22], involving the detection of EosFP–ATG8 processing upon induction of autophagy with AZD/ConA/E64d treatment. The assay demonstrated efficient processing of the fusion protein and accumulation of free EosFP in the plants expressing EosFP–ATG8E ΔC in the WT and *atg4a/b*, but not in the *atg5* background (Fig. 2E), thereby corroborating restored autophagic activity in the absence of ATG8 delipidation.

We hypothesized that lack of ATG8-delipidating activity would result in a large number of EosFP–ATG8E ΔC-decorated autophagosomes fusing with the vacuolar membrane, which should eventually yield fluorescent labeling of the tonoplast (Fig. 2F). Indeed, induction of autophagy caused accumulation of EosFP signal at the tonoplast in the *atg4a/b*, but not in the WT cells (Fig. 2G, H).

To confirm that the above-described observations were not specific to ATG8E, we generated transgenic lines expressing an artificially truncated version of another highly expressed Arabidopsis ATG8 isoform, ATG8F ΔC, in the WT and *atg4a/b* backgrounds. Indeed, treatment of these plants with AZD/ConA triggered a similar accumulation of autophagic bodies in both backgrounds, consistently observed in the roots of seedlings and true leaves of 4-week-old plants (Fig. 2I and Supplementary Fig. 4A, B). qPCR analysis of the established lines showed a slightly lower expression level of the transgene in the *atg4a/b* vs WT background (Supplementary Fig. 4C), which still yielded a comparable quantity of autophagic bodies (Supplementary Fig. 4B) and GFP-cleavage efficacy (Supplementary Fig. 4D) in both backgrounds. In line with our findings from ATG8E ΔC-expressing plants, increased levels of ATG8F–PE were observed in the *atg4a/b* background (Supplementary Fig. 4E), illustrating a defect in ATG8 delipidation.

We tested the reproducibility of the observed effect in an alternative ATG4-deficient background, specifically in the *atg4a-1/atg4b-1* knockout, which was utilized in the original article investigating the role of ATG8 delipidation in Arabidopsis autophagy[8]. For this, we extracted mesophyll protoplasts from the leaves of *atg4a-1/atg4b-1* mutants and the WT plants of the corresponding Wassilewskija-0 ecotype. The protoplasts were transformed to express GFP-tagged full-length or artificially truncated ATG8E or ATG8F isoforms, treated with AZD/ConA and imaged using confocal microscopy (Fig. 2J and Supplementary Fig. 5). We observed that the expression of artificially truncated ATG8 isoforms restored the accumulation of autophagic bodies also in this ATG4-deficient background.

Finally, to rule out the possibility of unforeseen artifacts associated with using artificially truncated ATG8s, we conducted an additional control experiment validating the functionality of such mutants. Both ATG8E ΔC and ATG8F ΔC variants were able to restore the accumulation of autophagic bodies in the vacuoles of the ATG8 loss-of-function mutant, indicating their sufficiency in reestablishing autophagic activity (Supplementary Fig. 6). We then focused on ATG8F ΔC-expressing lines for further analysis, as they were not prone to somatic silencing and were hence more suitable for the assays (Fig. 3). Using autophagic body quantification (Fig. 3A, B), a GFP-cleavage assay (Fig. 3C), and root growth phenotyping under nitrogen-depleted conditions (Fig. 3D, E), we confirmed that the artificially truncated ATG8 variant could effectively restore autophagic activity.

In summary, we found that while, as expected, the ATG4-deficient plants expressing artificially truncated ATG8 isoforms accumulate ATG8–PE adduct, these plants, unexpectedly, do not exhibit any defects in the formation or vacuolar delivery of autophagosomes.

### ATG8 delipidation-independent autophagy restores wild-type-like stress tolerance in the ATG4-deficient plants

While we confirmed the normal formation of plant autophagosomes in the absence of ATG8 delipidation, it still remained a question of whether such autophagosomes could carry out their intended function. To test this, we examined the stress tolerance of *atg4a/b* plants expressing EosFP-ATG8E ΔC under macronutrient-deprived conditions which would rely on autophagic activity[1,3] (Fig. 4A–F).

We performed uninterrupted time-lapse imaging of seedlings under nitrogen-depleted (−N) and carbon-depleted (−C) conditions. For this, seeds of WT, *atg4a/b* and *atg5* plants expressing EosFP-ATG8E ΔC were plated on standard 0.5× MS growth medium, and on growth media depleted of either nitrogen or sucrose. Plates were photographed every hour for 7 days using a SPIRO imaging platform[24]. For −C starvation induction, imaging was performed in the dark growth chamber using SPIRO green LED for illumination during image acquisition, to prevent photosynthetic carbon assimilation. We observed stagnated root growth in autophagy-deficient plants (*atg4a/b* and *atg5*) under both −N and −C conditions (illustrated in Fig. 4A–C; time-lapse movies showing complete plates are available in Supplementary Movies 2 and 3). Expression of EosFP-ATG8E ΔC in the WT background did not have a discernible effect on root growth, however, it reconstituted WT-like root growth in the *atg4a/b* background, revealing functional autophagic activity in these plants. No such effect was observed in the *atg5* plants expressing EosFP–ATG8E ΔC. These results were corroborated by our observations that the expression of another artificially truncated ATG8 isoform, ATG8F, reinstated the tolerance of the *atg4a/b* mutants to nitrogen depletion (Supplementary Fig. 4F, G).

Similar to root phenotyping, assessment of seedling shoot viability and leaf growth under the same stress conditions revealed restored stress tolerance in the *atg4a/b* upon expression of ATG8E ΔC (Fig. 4D–F). Uncropped images of plates with plants used in these experiments are shown in Supplementary Fig. 7.

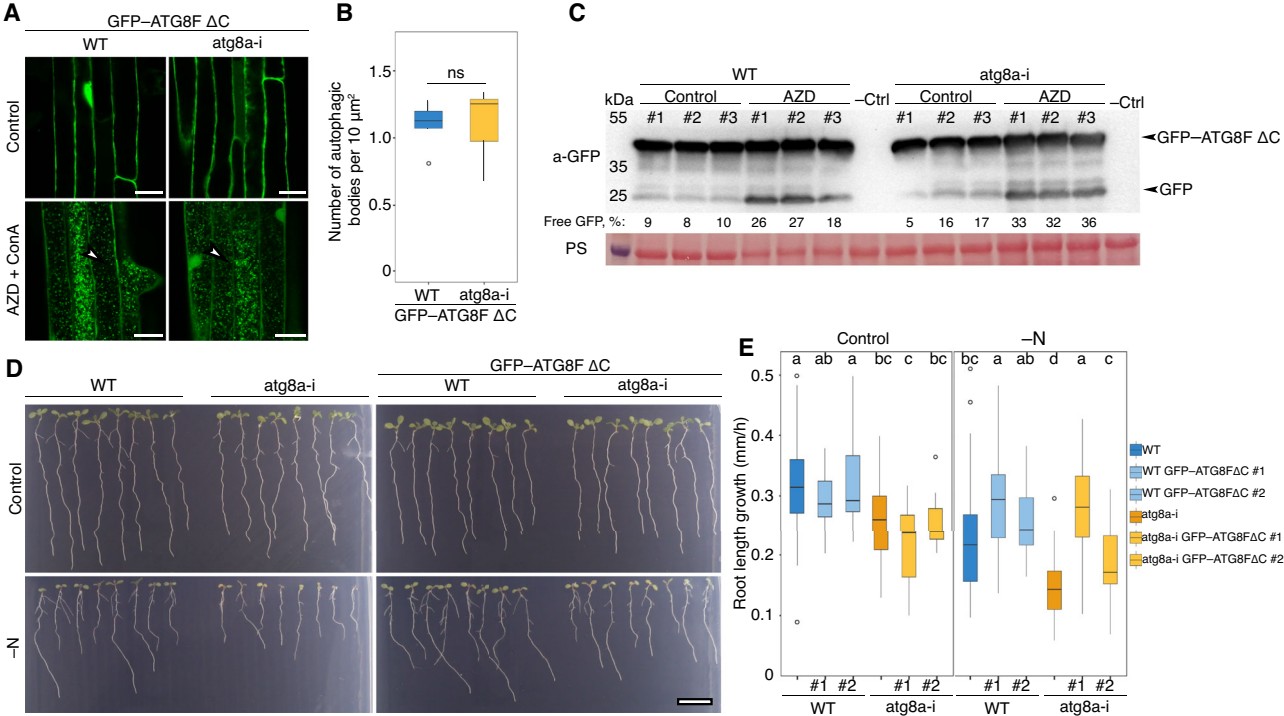

**Fig. 3 | Artificially truncated ATG8 restores autophagic activity in the ATG8 loss-of-function mutant. A** CLSM of Arabidopsis roots expressing GFP–ATG8FΔC in WT and *atg8a-i* backgrounds treated with AZD/ConA. White arrowheads indicate autophagic bodies. Scale bars, 20 μm. **B** Vacuolar puncta quantification for the samples (*n* = 12 biological replicates) illustrated in (**A**). **C** Western blot detection of GFP–ATG8FΔC cleavage in WT and *atg8a-i* backgrounds shows similar levels of free GFP accumulation in both WT and *atg8a-i* backgrounds upon induction of autophagy, suggesting that the introduction of GFP–ATG8FΔC restored autophagic flux in the ATG8 loss-of-function mutant. The numbers under corresponding lanes represent the integrated density of the free GFP band expressed as % of the total

signal intensity for the corresponding sample. PS Ponceau S, –Ctrl, protein extract from Arabidopsis seedlings not expressing GFP-tagged fusions. **D** Root phenotype of 11-day-old seedlings incubated for 6 days on control or nitrogen-depleted medium. Expression of the artificially truncated ATG8FΔC isoform is sufficient to alleviate the autophagy-deficient phenotype of *atg8a-i*. Seedlings were grown for 5 days on standard 0.5× MS medium, checked for GFP signal, and then transferred to plates with standard 0.5× MS (Control) or nitrogen-depleted (–N) medium to be imaged for 7 days. Scale bar, 1 cm. **E** Quantification of root growth on control and –N medium (*n* = 288 seedlings per treatment) in the WT and *atg8a-i* expressing the artificially truncated GFP–ATG8FΔC (illustrated in **D**).

Finally, we phenotyped the transgenic plants expressing artificially truncated ATG8E and ATG8F isoforms at the later developmental stages. Plants were grown in soil under long-day conditions to detect the early senescence phenotype typical for autophagy-deficient Arabidopsis plants[8]. Consistent with the above-described phenotyping assays, we could observe alleviation of the early senescence phenotype in the *atg4a/b* plants expressing either of the two truncated ATG8 isoforms (Fig. 4G, H and Supplementary Fig. 4H, I).

In summary, the phenotypic analyses of stress-tolerance of the photosynthetic organs and roots, carried throughout the life span of Arabidopsis, indicate that autophagic structures formed in the absence of ATG8 delipidation correspond to fully functional autophagosomes.

### Arabidopsis ATG8 isoforms differ in their capacity to carry out autophagic flux

Two out of nine Arabidopsis ATG8 isoforms, ATG8H and ATG8I, lack the C-terminal peptide and therefore do not require processing by ATG4 for their lipidation. It is conceivable that these two isoforms can be readily lipidated and used for biogenesis of autophagosomes in the *atg4a/b* mutant (Figs. 1A, B and 5A). Reflecting upon our discovery that autophagy can be re-established in the ATG4-deficient background through the expression of the artificially truncated ATG8E or ATG8F, we wondered why the natively truncated ATG8 isoforms do not similarly sustain autophagosome formation in *atg4a/b* plants. We considered two hypotheses: (i) the sum amount of ATG8H and ATG8I (which would correspond to the total amount of

ATG8 available for lipidation in *atg4a/b*) is much lower than the sum amount of all isoforms ATG8A–ATG8I available for lipidation in the WT background, and might not be sufficient for normal autophagosome biogenesis; (ii) ATG8H and ATG8I isoforms play specific roles in plant autophagy that do not allow efficient autophagosome biogenesis in the absence of other isoforms, e.g., sequestration of specific cargo. Notably, both ATG8H and ATG8I show higher protein sequence similarity with animal ATG8 orthologs compared to ATG8A–G (Supplementary Fig. 8A, B)[14], suggesting a potential functional diversification of these two groups of the Arabidopsis ATG8 isoforms.

To test our hypotheses, we selected ATG8F and ATG8I as the two strongest expressed ATG8 isoforms, one with and one without C-terminal peptide, respectively (Fig. 5B). We generated fluorescently labeled fusions of these isoforms and expressed them transiently from a strong native Arabidopsis promoter (*APA1*)[25] in WT, *atg4a/b*, and *atg7* cells, using the latter as a negative control lacking ATG8 lipidation activity. Upon induction of autophagy, we could observe the expected accumulation of autophagic-body-like structures in the vacuoles of WT cells expressing either of the ATG8 isoforms, and lack thereof in the vacuoles of *atg7* mutants (Supplementary Fig. 8C). Intriguingly, over-expression of ATG8I in the *atg4a/b* background seemed to have reconstituted accumulation of puncta in this background, thus supporting our first hypothesis that the amount of ATG8 available for lipidation might be the limiting factor in *atg4a/b* autophagy. As expected, ATG8F overexpression in the same cells did not reconstitute autophagosome biogenesis.

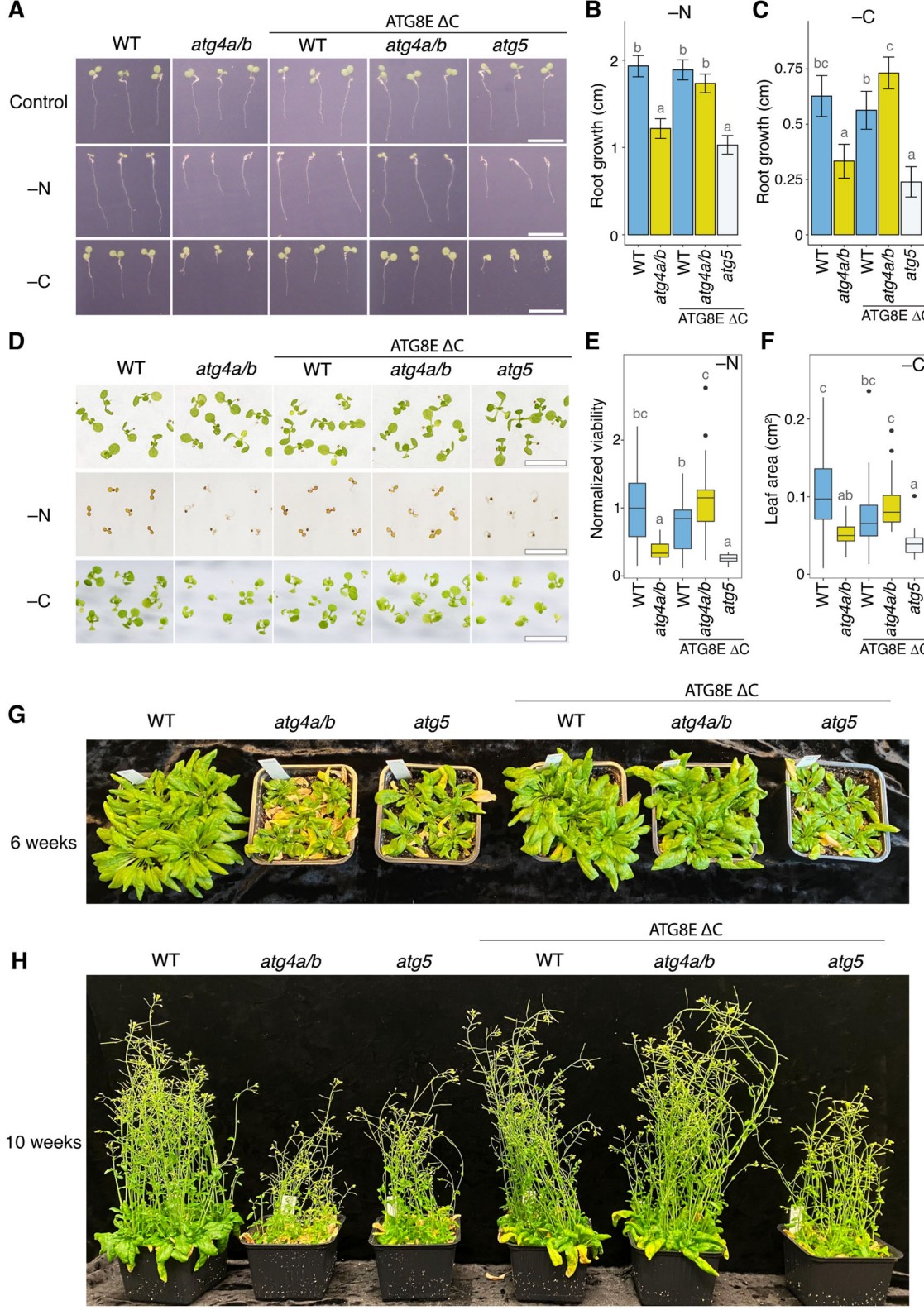

To enable an in-depth investigation of this finding using biochemistry and phenotyping assays, we generated stable transgenic lines overexpressing GFP-fusions of the same two ATG8 isoforms under the control of a strong viral promoter (2×35S) and verified expression levels of the transgenes in the new lines using qPCR (Supplementary Figs. 4C and S8D). Detection of autophagic bodies in the true leaves of these plants confirmed the results obtained in the transient expression system: indeed, overexpression of ATG8I was sufficient to reconstitute accumulation of autophagic bodies in the vacuoles of *atg4a/b* upon AZD/ConA treatment (Fig. 5C). The GFP-cleavage assay performed on these transgenic lines further confirmed autophagic flux present in the *atg4a/b* overexpressing the natively truncated ATG8I isoform (Fig. 5D). Intriguingly, detection of the lipidated and non-lipidated adducts of ATG8 in the protein extracts from plants overexpressing ATG8F and

**Fig. 4 | Expression of ATG8EΔC alleviates autophagy-deficient phenotypes of *atg4a/b*. A** Examples of Arabidopsis seedlings phenotyped using SPIRO under control, nitrogen-depleted (−N), and carbon-depleted (−C) conditions. Normal root growth under nutrient-depleted conditions is restored in *atg4a/b* seedlings expressing ATG8EΔC. In total, 997 biological replicates were phenotyped in seven independent experiments. Scale bars, 1 cm. **B** Quantification of seedling root length on the 4th day after germination under −N conditions, as illustrated in (**A**). The chart represents combined data from two independent experiments, *n* = 364 seedlings. Error bars indicate a 95% confidence interval, a complete analysis description is provided in "Methods". **C** Quantification of seedling root growth during four days of recovery following four days of carbon depletion, as illustrated in (**A**). The chart represents combined data from two independent experiments, *n* = 283 seedlings. Root length was analyzed and visualized as in (**B**). **D** Examples of Arabidopsis seedlings shoot phenotypes under control, −N and −C conditions. Scale bars, 1 cm.

Photos of complete plates are in Supplementary Fig. 8. **E** Viability of Arabidopsis seedlings grown on −N medium illustrated in (**D**) was estimated by assessing shoot chlorosis. The chart shows representative data from one out of two independent experiments, *n* = 157 seedlings. A complete analysis description is provided in Methods. **F** Efficacy of recovery after carbon depletion illustrated in (**D**) was estimated by measuring the leaf area for each seedling at the end of a one-week-long recovery after 7 days of carbon depletion. The chart shows representative data from one out of two independent experiments, *n* = 160 seedlings. Results were analyzed and visualized as in (**E**). **G** Rosette phenotype of 6-week-old plants grown under long day conditions show a typical attribute for autophagy deficiency: an early onset of senescence in the *atg4a/b* and *atg5* backgrounds, and lack thereof in the WT and *atg4a/b* plants expressing ATG8EΔC protein. **H** Inflorescence phenotype was imaged one month later for the same plants as shown in (**G**).

ATG8I isoforms revealed accumulation of ATG8I–PE not only in the *atg4a/b* but also in the WT background (Fig. 5E), suggesting a decreased affinity of ATG4 towards this isoform.

However, although the GFP-cleavage assay revealed the occurrence of autophagic activity in the ATG4-deficient plants overexpressing ATG8I, the decreased cleavage of GFP–ATG8I in the *atg4a/b* background in comparison to the WT (Fig. 5D, E) suggested that our second hypothesis might also be correct, and that ATG8I cannot fully reconstitute normal autophagosome formation. To verify this further, we compared the number of puncta formed in the Arabidopsis root cells overexpressing ATG8E ΔC, ATG8F, or ATG8I upon upregulation of autophagy. For this, Arabidopsis seedlings overexpressing either of the ATG8 isoforms in the WT, *atg4a/b*, or *atg5* backgrounds were subjected to AZD/ConA treatment for 24 h followed by confocal microscopy (Fig. 5F). Quantification of the puncta on the obtained micrographs revealed no significant difference between puncta density in the WT and *atg4a/b* overexpressing ATG8I (Fig. 5G), indicating normal rate of autophagosome formation and their delivery to the vacuole. However, it did not escape our notice that the size of the GFP–ATG8I-positive puncta differed in the WT and *atg4a/b* cells. Quantification of the puncta size distribution showed that indeed, autophagic bodies formed in the *atg4a/b* cells overexpressing ATG8I were significantly smaller in comparison to autophagic bodies formed in the vacuoles of WT cells overexpressing the same isoform (Fig. 5H), indicating that autophagosome elongation could be compromised under conditions where ATG8I predominates, with possible presence of minor amounts of ATG8H.

Next, we tested if the decreased size of autophagic bodies in the ATG8I-overexpressing *atg4a/b* plants impacted the efficacy of autophagic activity. We deprived seedlings of nitrogen and measured the elongation efficacy of their roots using SPIRO time-lapse images (Fig. 5I, J). Notably, the overexpression of the natively truncated ATG8I did not rescue the stagnated root growth phenotype of *atg4a/b*, indicating that smaller autophagosomes observed in this background were not sufficient to cope with the scarcity of exogenous nitrogen. Surprisingly, under the same conditions, we also observed decreased root elongation in the WT plants overexpressing ATG8I (Fig. 5I, J).

Considering that the puncta quantification experiments (Fig. 5F, G) were performed using AZD as a trigger for autophagy, while root phenotyping was done under −N conditions (Fig. 5I, J), we decided to test if the overexpression of ATG8I has a different effect on WT depending on the stress conditions. For this, we compared the density and size of autophagic bodies upon AZD, −N, and −C treatments (Supplementary Fig. 9) in the lines overexpressing either ATG8E ΔC, which caused no discernible phenotype in the WT in the previous experiments, or ATG8I which caused decreased tolerance to nitrogen depletion in WT. In agreement with our previous observations, ATG8E ΔC overexpression in the WT and *atg4a/b* backgrounds yielded similar density and size of autophagic bodies under all three stress conditions, while over-expression of ATG8I resulted in smaller autophagic bodies (Supplementary Fig. 9A–C). The latter phenotype was especially prominent

under −N conditions (for WT: 0.51 µm$^2$ under −N vs 0.76 µm$^2$ under −C, or 0.69 µm$^2$ under AZD, Tukey's HSD test *p*-value < 0.001; Supplementary Fig. 9A–C), thereby explaining the observed stunted root growth on nitrogen-depleted medium. The autophagic flux rate under the three stress conditions was additionally confirmed using the GFP-cleavage assay (Supplementary Fig. 9D). Since ATG8I overexpression reduced the size of autophagic bodies in the WT cells under all three types of stress conditions, it is tempting to speculate that an excessive amount of this isoform could cause premature closure of phagophores.

Interestingly, when plants were grown under favorable long-day conditions in the soil, WT overexpressing ATG8I showed no discernible phenotype, while *atg4a/b* plants overexpressing the same isoform grew fitter than autophagy-deficient knockouts (Fig. 5K). These observations suggest that ATG8I overexpression can rescue autophagy of the *atg4a/b* knockout to a level that is sufficient for WT-like plant fitness under favorable conditions, and that deleterious effects of ATG8I overexpression in the WT is specific for certain types of stresses, including −N.

In summary, we demonstrated that the autophagy-deficient phenotype of *atg4a/b* is caused by the insufficient amount of the natively truncated ATG8 isoforms available for lipidation during autophagosome formation. We showed that overexpression of the natively truncated ATG8I is sufficient to reconstitute the normal rate of autophagosome formation and their delivery to the vacuole but yields autophagosomes almost twice smaller than those formed when artificially truncated ATG8E isoform is expressed in the same background. Furthermore, overexpression of the natively truncated ATG8I isoform undermines the tolerance of WT plants to nitrogen deprivation, possibly by shifting autophagy selectivity to the one unfavorable under such conditions.

## Ultrastructural analysis confirms that vacuolar puncta observed in ATG4-deficient plants expressing truncated ATG8 isoforms indeed represent autophagic bodies

To further verify the nature of the vacuolar puncta accumulating in the Arabidopsis transgenic lines upon induction of autophagy, we performed transmission electron microscopy (TEM). For this, seedlings of the transgenic lines expressing the artificially truncated ATG8E ΔC, the natively truncated ATG8I, and full-length ATG8F isoforms in the WT and *atg4a/b* backgrounds were subjected to AZD/ConA treatment followed first by confocal laser scanning microscopy (CLSM) and later by TEM (Fig. 6). Predictably, for the *atg4a/b* plants, CLSM revealed the accumulation of the fluorescent puncta only in the vacuoles of lines expressing truncated versions of ATG8s, whereas vacuolar puncta could be detected for all lines in the WT background. TEM showed that the vacuoles of ATG8E ΔC-expressing lines accumulate numerous vesicles filled with varied cargo, including portions of cytoplasm and organelles. No such vesicles were detectable in the *atg4a/b* cells expressing full-length ATG8F. Notably, vacuolar vesicles observed in the *atg4a/b* cells expressing natively truncated ATG8I were noticeably

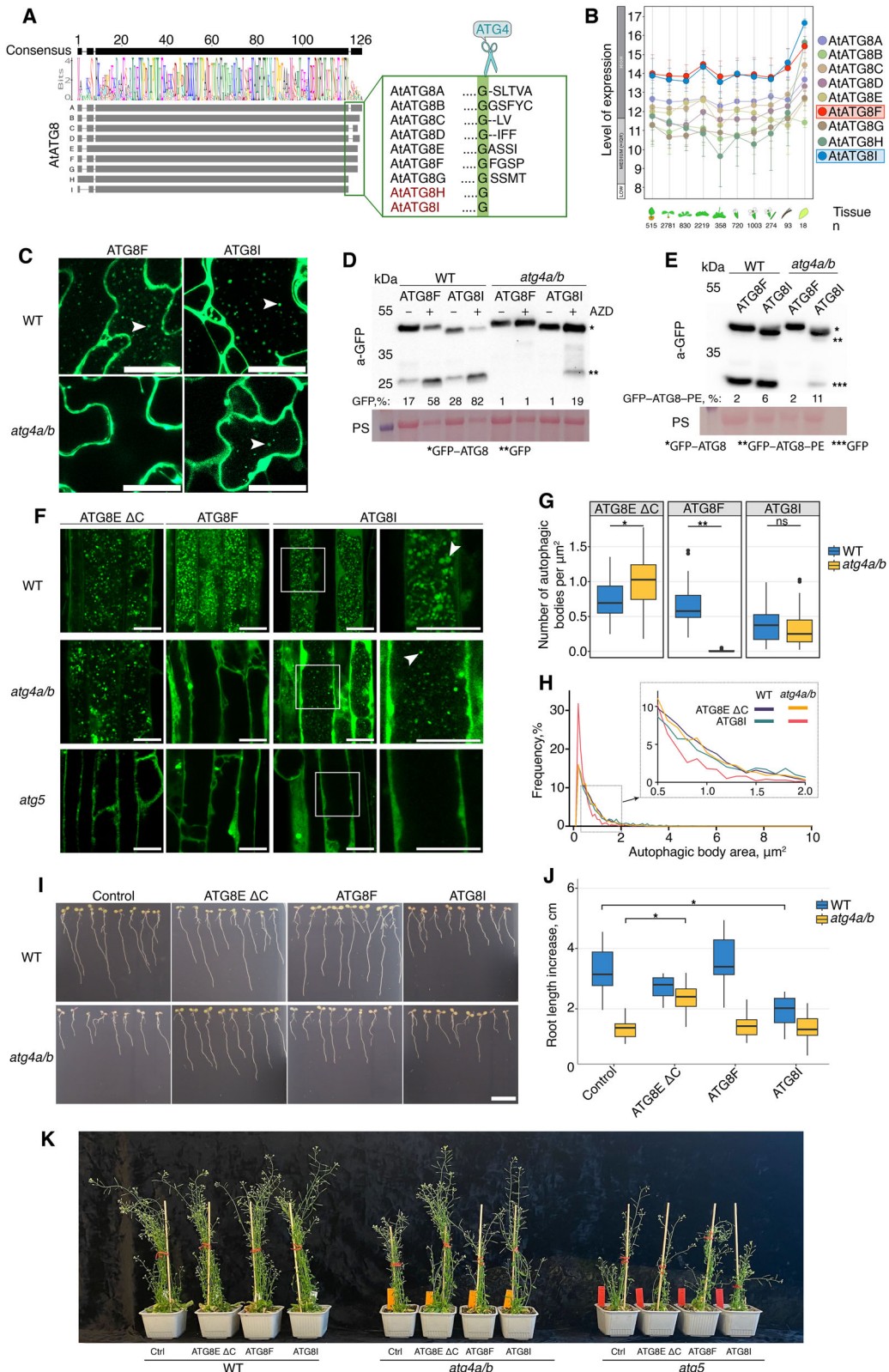

smaller than those observed in the vacuoles of ATG8E ΔC-expressing cells (Fig. 6). These findings further support the quantitative data obtained through high-throughput CLSM-based assessment of autophagic body size (Supplementary Fig. 9A–C).

In summary, ultrastructural analysis of the vacuolar puncta observed in the transgenic lines confirmed that they represent autophagic bodies.

## ATG8 delipidation is critical for autophagy in a green microalga *C. reinhardtii*

The current phylogeny categorizes green plants (*Viridiplantae*) as a monophyletic taxon consisting of two evolutionary lineages that diverged over a billion years ago: the Chlorophyta (comprising some of the green algae) and Streptophyta (including both land plants and remaining green alga)[26,27]. To investigate whether independence of

**Fig. 5 | Arabidopsis ATG8 isoforms differ in their capacity to carry out autophagic flux. A** Schematic representation of Arabidopsis ATG8 isoform alignment. The inset displays the C-termini with the Gly residue critical for lipidation highlighted in green. Notably, ATG8H and ATG8I lack the C-terminal peptide, making them natively truncated isoforms with exposed critical Gly. **B** GENEVESTIGATOR data on organ-specific expression of all nine ATG8 isoforms of Arabidopsis. ATG8F and ATG8I isoforms are expressed at the highest level in all tissues. **C** CLSM of Arabidopsis leaves expressing GFP-tagged ATG8F or ATG8I in WT or *atg4a/b* treated with AZD/ConA. White arrowheads indicate autophagic bodies. Scale bars, 25 μm. **D** GFP-cleavage assay performed on the transgenic lines illustrated in (**C**), showing a restored accumulation of free GFP in the *atg4a/b* plants expressing GFP–ATG8I. Densitometry results are shown under corresponding lanes. PS, Ponceau S. **E** ATG8I-PE is detectable in the samples shown in (**D**), when separated on the gel containing 6 M urea. Densitometry results are shown under corresponding lanes. PS, Ponceau S. **F** CLSM of Arabidopsis roots expressing

ATG8EΔC, ATG8F or ATG8I in the WT, *ATG4-* and *ATG5*-knockouts treated with AZD/ConA. Scale bars, 20 μm. White arrowheads indicate autophagic bodies. **G** Vacuolar puncta quantification for samples illustrated in (**F**). Welch's *t*-test, $n = 67$ biological replicates. *$p < 0.05$; **$p < 0.01$; ns not significant. **H** Size distribution of autophagic bodies in the samples illustrated in (**F**). The inset highlights the increased frequency of smaller autophagic bodies in the *atg4a/b* plants expressing ATG8I. **I** Root growth phenotype −N medium. Unlike ATG8EΔC, the natively truncated ATG8I isoform is not sufficient to alleviate the autophagy-deficient phenotype of *atg4a/b*. Scale bar, 1 cm. **J** Quantification of the increase in root length of the seedlings after 7 days of growth on the −N medium (illustrated in **I**). Individual Dunnett's tests for WT and *atg4a/b*, $n = 128$ seedlings. *$p < 0.0001$. **K** Representative phenotypes of two-month-old plants. Ctrl, a not transformed plant of the corresponding genotype; ATG8EΔC, ATG8F, and ATG8I, overexpression of the fluorescent fusion of the corresponding ATG8 isoform.

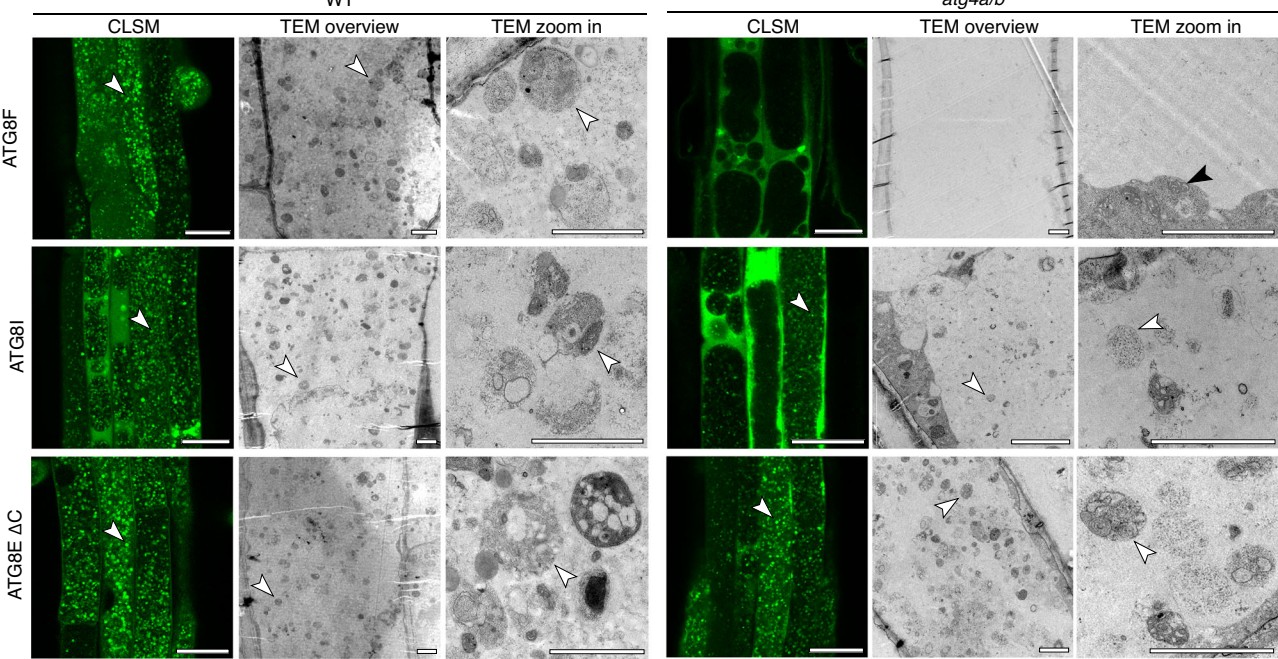

**Fig. 6 | Ultrastructure of GFP-positive puncta accumulating in the vacuoles upon expression of full-length and truncated GFP-tagged ATG8 isoforms.** Seven-day-old Arabidopsis seedlings expressing GFP-tagged full-length (ATG8F), natively truncated (ATG8I) or artificially truncated (ATG8EΔC) ATG8 variants in WT or *atg4a/b* backgrounds were subjected to AZD/ConA treatment for 2 h prior to confocal imaging (CLSM). After CLSM, seedlings were fixed and processed for transmission electron microscopy (TEM). Ultrastructural analysis of the GFP-positive puncta detected with CLSM revealed numerous autophagic bodies with various cargo (cytoplasm, mitochondria, ER). No autophagic bodies were detected

in the vacuoles devoid of GFP-positive puncta (*atg4a/b* plants expressing GFP–ATG8F). In agreement with our observations made using CLSM (Fig. 5H), autophagic bodies detectable in the *atg4a/b* plants expressing natively truncated GFP–ATG8I isoform were visibly smaller than in the *atg4a/b* plants expressing artificially truncated GFP–ATG8EΔC or in WT expressing any of the checked ATG8 isoforms. White arrowheads point at autophagic bodies containing organelles and portions of cytoplasm. The black arrowhead points at mitochondria in the cytoplasm. Scale bar for CLSM images, 25 μm. Scale bars for all TEM micrographs, 2 μm.

---

autophagy on ATG8 delipidation is inherent for green plants we conducted corresponding experiments on the unicellular green alga, *C. reinhardtii* (hereafter referred to as Chlamydomonas). Most interestingly, in-depth studies of these algae revealed that they possess numerous animal-like functions[28,29]. The Chlamydomonas genome encodes single genes for the core ATGs, making it a much simpler model for autophagy research compared to Arabidopsis. Intriguingly, the sole ATG8 protein of Chlamydomonas retained its C-terminal peptide, which necessitates the ATG4-mediated processing prior to ATG8 lipidation (Fig. 1A, B). Therefore, we generated a truncated version of Chlamydomonas ATG8, CrATG8 ΔC, which had the critical C-terminal Gly exposed and could be lipidated in the absence of ATG4 activity but would depend on it for the delipidation.

First, we generated *cratg4* and *cratg5* autophagy-deficient mutants in the commonly used UVM4 background (Supplementary

Fig. 10) and complemented them with mCherry-tagged full-length CrATG8 or CrATG8 ΔC. Immunoblot analysis showed a good expression level of both transgenes in the mutant strains (Fig. 7A). A lack of autophagy elicits an early senescence phenotype, especially under nutrient-scarce conditions[30,31]. Therefore, we assessed the functionality of the autophagic pathway in the established strains by assessing their rate of senescence. We cultivated the strains in the standard TAP growth medium for 40 days and monitored the degree of their chlorosis while they were gradually depleting nutrients from the medium (Fig. 7B). Expectedly, *cratg5* mutants complemented with full-length or truncated ATG8 but still deficient in ATG8 lipidation exhibited chlorosis associated with the early senescence phenotype, while WT cells remained dark green under the same conditions (Fig. 7B). Notably, the early onset of chlorosis in the *cratg4* mutant was not rescued by the expression of the truncated ATG8 version (Fig. 7B).

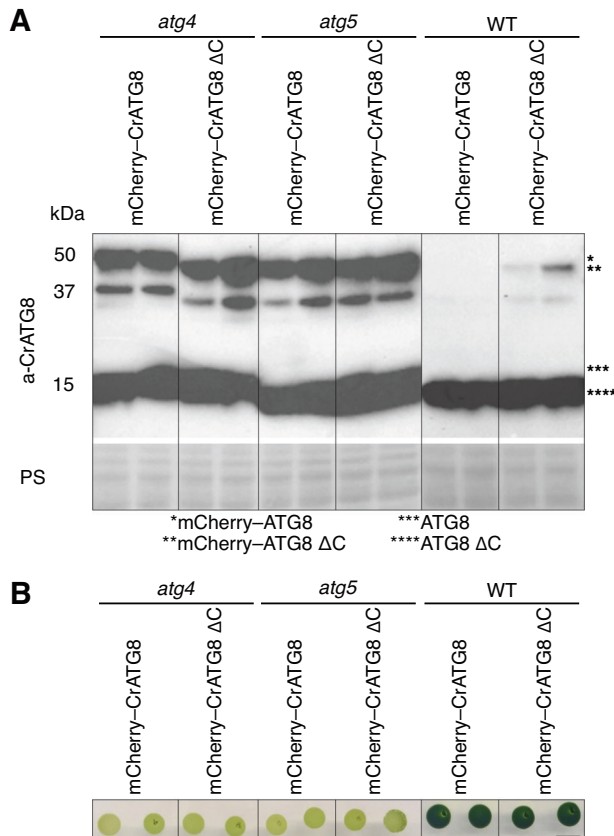

**Fig. 7 | Delipidation of ATG8 is critical for autophagy in *C. reinhardtii*. A** Western blot detection of endogenous ATG8 and mCherry−CrATG8 transgene expression in the WT, *cratg4*, and *cratg5* backgrounds. Ponceau S staining was used as a loading control. The cells were grown in the liquid TAP medium under standard conditions and harvested at a density of $1-2 \times 10^7$ cells ml$^{-1}$. **B** Senescence phenotype of the transgenic *Chlamydomonas* lines grown on the TAP medium for 40 days. The light color of colonies indicates nutrient deficiency sensitivity caused by dysfunctional autophagy. Scale bar, 1 cm.

Taken together, our data suggests that, unlike in the vascular plant, Arabidopsis, but similar to yeast and mammals, the delipidation of ATG8 plays a crucial role in the autophagy of Chlamydomonas.

## Discussion

In this study, we demonstrate that ATG8 delipidation is dispensable for autophagy in the vascular plant *A. thaliana*, which belongs to the Streptophyta lineage. However, it is of crucial importance for autophagy in the unicellular green alga *C. reinhardtii*, which is part of the Chlorophyta lineage and has preserved a number of animal-like functions[28,29]. The specific evolutionary adaptations that rendered delipidation dispensable have yet to be uncovered.

Since ATG8 delipidation was previously reported to be crucial for autophagy in plants[8], as well as in yeast and animal cells[6,10,11], we considered it imperative to conduct a thorough verification of our surprising observation regarding the reconstituted autophagosome formation in the ATG4-deficient Arabidopsis. We employed a combination of mutagenesis, cell biology, biochemistry, and phenotyping assays to establish a robust foundation for future investigations aimed at elucidating the molecular mechanisms governing plant-specific maturation of autophagosomes possible in the absence of ATG8 delipidation. Furthermore, we validated our observations using two artificially truncated ATG8 isoforms (ATG8E ΔC and ATG8F ΔC) in two independent ATG4-loss-of-function mutants (*atg4a-1/atg4b-1* and *atg4a-2/atg4b-2*, the former of which

was used in the original study about the role of ATG8 delipidation in Arabidopsis[8]).

Importantly, upon thorough evaluation, we determined that none of our results directly contradicted data presented in the study by Yoshimoto and colleagues[8]. The differences in our conclusions probably stem from serendipitous factors. For example, in our study, we elected to test artificially truncated ATG8E and ATG8F isoforms which unexpectedly turned out to be more potent in enabling autophagic flux than the natively truncated ATG8I used by Yoshimoto and colleagues. Furthermore, as our results demonstrate, overexpression of GFP−ATG8I yields a lower autophagic activity in *atg4a/b* than in WT. Therefore, it is conceivable that the activity in *atg4a/b* might have fallen below the detection limit for the assay relying on the imaging of the diffuse GFP signal diluted in the vacuolar volume, implemented by Yoshimoto and colleagues. In our study we opted for imaging autophagic bodies represented by bright puncta of concentrated GFP signal, thereby increasing the chance of detecting even weak autophagic activity.

Based on our observations, we propose that plant autophagosome maturation diversified into strategies dependent on and independent from ATG4 delipidating activity (Fig. 8). It is an intriguing question, how a critical step regulating autophagosome maturation became unnecessary in some eukaryotes. Noteworthily, the concept of "conserved step" in this context should be viewed with some caution: while disruption of ATG8 delipidation indeed halts autophagosome biogenesis in yeast and mammalian cells, it affects different stages of the process in these organisms[6,10,12,13], indicating inherent differences in why this step is critical for autophagy in various eukaryotes. Additionally, more recent studies suggest that the dependency of mammalian autophagy on ATG8 delipidation might be model/condition-specific[21,32].

Autophagy was likely invented by the last eukaryotic common ancestor (LECA)[33] to maintain the newly acquired complex cell structures, such as organelles. During the evolution of eukaryotes, autophagy has diversified to adapt to the various life strategies, mobility traits, nutrient requirements, and specific configurations of endomembrane trafficking systems observed in fungi, animals, plants, and algae. For example, eukaryotes evolved different configurations of the lytic compartment−the final destination of the autophagic pathway. The main lytic compartments of animal cells are compact and mobile lysosomes. Upon the induction of autophagy, both lysosomes and autophagosomes are trafficked toward their fusion location (Fig. 8). The modest size of lysosomes and their importance for other cellular pathways compels animal cells to regenerate these organelles after their fusion with autophagosomes, in a process known as autolysosome recovery (ALR)[34]. Such a process has not yet been reported for fungal and plant cells, in which autophagosomes fuse with the large lytic vacuoles that often occupy most of the cell volume[1,3].

Notably, autophagosome biogenesis in yeast cells occurs in proximity to the vacuole, at the single phagophore assembly site[2] (Fig. 8). Consequently, once complete, autophagosomes fuse immediately with vacuoles. In contrast, autophagosomes in animal cells emerge at multiple locations, and are trafficked towards nuclei to fuse there with the perinuclearly translocated lysosomes[35]. Interestingly, plants' autophagic strategy seems to be somewhere in between the yeast and animal solutions (Fig. 8). That is, autophagosomes form at multiple foci and are trafficked toward the immobile large lytic vacuole[3]. In view of these facts, it is conceivable that despite the conservation of the core autophagic mechanisms, some molecular aspects of the pathway might significantly differ in vascular plants, different groups of algae, fungi, and animals.

ATG8 recruits other core ATG proteins to the growing phagophore. Subsequently, delipidation of ATG8 from maturing autophagosomes was suggested to release PAS proteins prior to the fusion of the autophagosome with the lytic compartment, thus preventing their

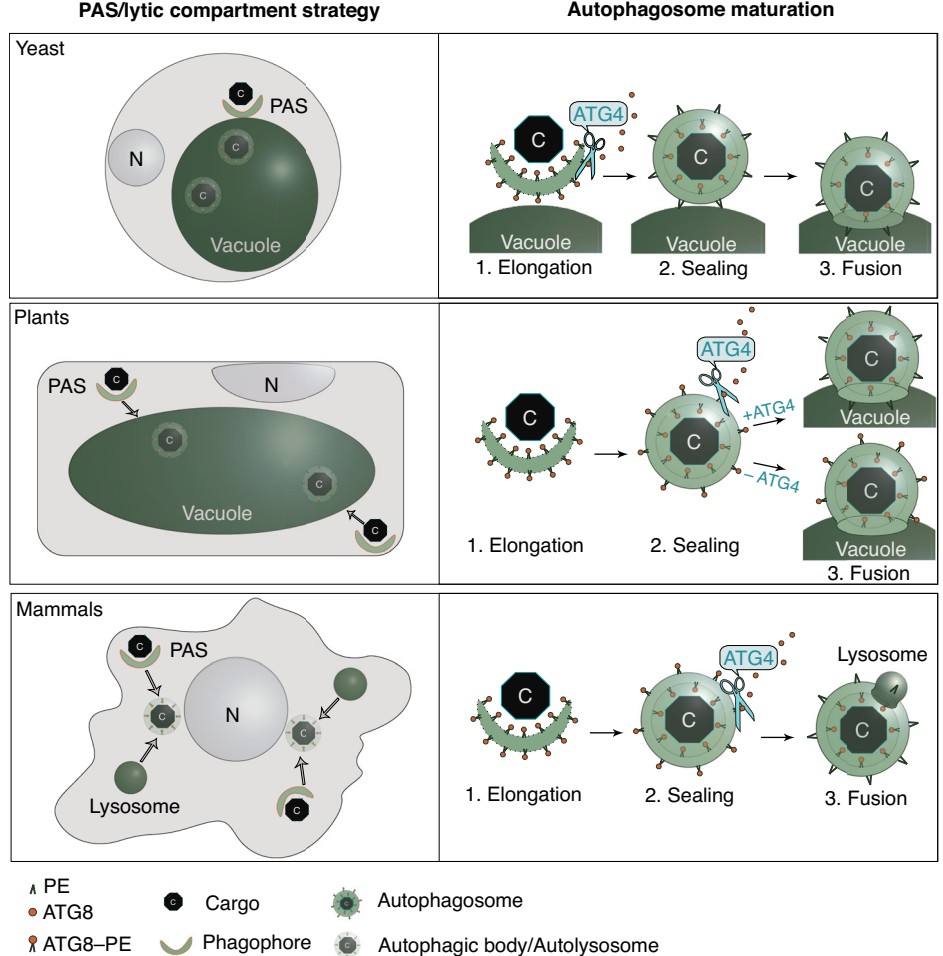

**Fig. 8 | Schematic representation of ATG8 delipidation importance for autophagy in the different eukaryotic kingdoms.** ATG4-dependent delipidation of ATG8 is considered to be generally conserved in plants, animals, and fungi. However, this process is critical for the early steps of autophagosome maturation in yeast cells (phagophore elongation), while in mammalian cells the same delipidation step is critical for the later stage of autophagy (docking of complete autophagosomes to lysosomes). In this study, we show that although ATG8 delipidation occurs in plants, it is dispensable for autophagy. The different roles ATG8-delipidation plays in autophagosome maturation might be representative of autophagy adaptation to the various strategies for the lytic compartment (single large immobile vacuoles in plant and yeast cells vs numerous small motile lysosomes in animal cells), for phagophore assembly sites (single site in yeast vs numerous sites in animal and plant cells), and autophagosome trafficking towards the lytic compartment (autophagosome formation in the proximity to the vacuole in yeast, trafficking of autophagosomes towards immobile vacuole in plants, coordinated trafficking of autophagosomes and lysosomes to their meeting spot in mammalian cells). C cargo, N nucleus, PE phosphatidylethanolamine.

unnecessary degradation[2]. Interestingly, such rescue of the core ATGs might be less critical for Arabidopsis autophagy. For instance, ATG1a was shown to be delivered to the vacuole together with autophagosomes[36]. Future studies identifying proteins decorating mature plant autophagosomes will help elucidate how they are affected by ATG8 delipidation.

In higher eukaryotes, some of the ATG gene families expanded quite significantly[3,14,37]. For example, Arabidopsis has nine ATG8 and two ATG4 orthologs. The reasons behind such expansion are not fully understood, but it is likely that functional diversification of ATG isoforms became necessary within the context of complex organisms. For instance, Zess and colleagues previously suggested a model according to which plant autophagosomes might vary in their ATG8 isoform content and consequently contain different cargo[38]. In the present study, we demonstrate drastic differences in the size of the autophagosomes formed with the participation of ATG8E and ATG8I isoforms, as well as in their impact on plant stress tolerance. In the future, we plan to determine the differences in the content of the ATG8F- and ATG8I-specific autophagic bodies and explain their impact on plant stress tolerance.

Current evidence indicates that Arabidopsis ATG4 orthologs are redundant: Arabidopsis knockouts of single *ATG4* genes do not have autophagy-deficient phenotypes[8,23](Supplementary Fig. 3I, J). Furthermore, we demonstrated that either of the ATG4 proteases is sufficient for the complete processing of ATG8 *in planta* (Supplementary Fig. 2E, F). Intriguingly, we also revealed that efficient delipidation of ATG8 requires the presence of both ATG4 proteases (Supplementary Fig. 3K, L), indicating that the delipidating activity of ATG4s is significantly less efficient than their proteolytic activity, which might play a role in preserving the pool of ATG8−PE required during phagophore expansion. Similarly, highly efficient proteolytic activity and slower delipidation activity were reported for the mammalian ATG4s[12].

Interestingly, yeast ATG4 contains the N-terminal motif, APEAR, responsible for its association with ATG8−PE during the delipidation step[9], while mammalian ATG4s were shown to associate with ATG8−PE via their C-terminal LIR motif[12]. It remains to be determined which of the predicted motives of Arabidopsis ATG4s is responsible for association with ATG8−PE. Yet, the APEAR motif is an unlikely candidate for it, as it is present only in the ATG4B, while both ATG4A and ATG4B partake in ATG8 delipidation (Supplementary Fig. 3K, L).

Here we report an additional intriguing observation of naturally truncated ATG8I isoform accumulating in its ATG8I–PE adduct in the WT upon upregulation of autophagy (Fig. 5E). This is quite uncharacteristic for plant ATG8s, as their lipidated adducts are notoriously difficult to detect in plant protein extracts[23]. Future studies will determine whether such accumulation is caused by the decreased affinity of ATG4 towards ATG8I. Such a loss of affinity is plausible because the natively truncated ATG8 isoforms do not require interaction with ATG4 for processing before lipidation, and their interaction with ATG4 for delipidation is dispensable for autophagy.

In conclusion, we discovered that in contrast to yeast, mammals, and Chlamydomonas, vascular plants do not require ATG8 delipidation for autophagosome maturation and delivery to a lytic compartment. In addition, our study provides novel insights into the isoform-specific role of the ATG8 family in plant autophagy and lays the foundation for future research on the mechanism of plant autophagosome maturation.

## Methods

### Arabidopsis plant material and growth conditions
Unless stated otherwise, all experiments were performed in Col-0 Arabidopsis accession (N1092). Autophagy-deficient mutants were described previously: *atg4a-2/atg4b-2*[23] (*atg4a/b* in this study), *atg5-1*[18](*atg5* in this study), *atg7-2*[39] (*atg7* in this study). WT and ATG4-deficient mutant (*atg4a-1/b-1*)[8] in Wassilewskija-0 accession (N1602) were kindly provided by Prof. D. Hofius. The nonuple *atg8a-i* knockout was kindly provided by Dr. Y. Dagdas group[40].

For cultivation on plates, Arabidopsis seeds were surface sterilized with bleach. In brief, seeds were agitated for 20 min in 10% *v/v* bleach (Klorin, Sweden) water solution supplemented with 0.05% Tween-20, washed in four changes of sterile MQ water and plated on 0.5× MS medium, comprising 0.5× MS (*Duchefa*, ref. M0222); 10 mM MES (*Duchefa*, ref. M1503); 1% sucrose; pH 5.8; 0.8% plant agar (*Duchefa*, ref. P1001). Plates were sealed with plastic wrap and incubated vertically under long-day conditions: 16/8-h light /dark cycle at PAR 150 μmol photons m$^{-2}$ s$^{-1}$, at 22 °C.

S-Jord soil (Hasselfors, Sweden) was used to grow Arabidopsis plants in pots. Plants were incubated under long-day conditions: 16/8-h light /dark cycle at PAR 150 μmol photons m$^{-2}$ s$^{-1}$, at 22 °C.

### Chlamydomonas strains and growth conditions
UV mutagenesis strain 4 (UVM4) capable of efficiently overexpressing foreign genes[41] was used as the WT background in this study. The *Chlamydomonas* cells were grown in tris-acetate-phosphate (TAP) medium at 23 °C and a light-dark cycle (LD16:8). The cells were harvested at a density of 1–2 × 10$^7$ cells ml$^{-1}$ for immunoblot analysis or genomic DNA isolation.

### Cloning of constructs
Constructs for studies in Arabidopsis were generated using GreenGate[42] and Gateway (Invitrogen) cloning systems. Construct coding N-terminal GST fusions were created for the expression of recombinant wild-type and proteolytically dead (PD) Arabidopsis ATG4 proteases. For this, ATG4A (AT2G44140) and ATG4B (AT3G59950) CDS were lifted from the total cDNA of *A. thaliana* using primers AM 417/418 and AM 419/420, respectively (Supplementary Table 1). Active sites of ATG4A and ATG4B were mutated using Quik-Change II Site-Directed Mutagenesis Kit (Agilent, 200523) and primer pairs AM 622/AM 623 and MA 624/625, respectively (Supplementary Table 1). Amplicons were recombined with pDONR/Zeo vector using the Gateway cloning system (Invitrogen). The resulting constructs AM 753 and AM 754 (Supplementary Table 2) further recombined with pDEST15 (Gateway™, Invitrogen) to obtain AM 810-813 destination constructs (Supplementary Table 2).

The constructs encoding EosFP–AtATG8E variants for expression in plants were generated using the Gateway cloning system (Invitrogen). To clone full-length ATG8E, ATG8E G/A, and ATG8E ΔC, the genomic sequence of ATG8E (AT2G45170) was PCR-amplified from the total genomic DNA of *A. thaliana* using primers AM 475/476, AM 475/478 and AM 475/477, respectively (Supplementary Table 1). Amplicons were recombined with pDONR/Zeo vector using the Gateway cloning system (Invitrogen). The resulting constructs AM 655-657 (Supplementary Table 2) were further recombined with pUBN EosFP[43] to obtain AM 661-663 destination clones (Supplementary Table 2).

The construct encoding EosFP–AtATG8A–CFP for expression in plants was generated by first fusing the ATG8A gene with CFP CDS using overlay PCR. For this, the ATG8A gene (AT4G21980) was lifted from the total genomic DNA of *A. thaliana* using primers AM 429/430, and the CDS of CFP was amplified from pGWB 661[44] (Supplementary Table 1). The obtained amplicons were fused in a PCR with primers AM 431/AM 428 (Supplementary Table 1) and the product was recombined with the pDONR/Zeo vector using the Gateway cloning system (Invitrogen). The resulting construct AM 556 (Supplementary Table 2) was further recombined with pUBN EosFP[43] to obtain the AM 567 destination clone (Supplementary Table 2).

The constructs encoding GFP–ATG8E ΔC, GFP–AtATG8F, GFP–AtATG8F ΔC, and GFP–AtATG8I fusion proteins for expression in plants were generated using the GreenGate system, following the published instructions[42]. pGGC ATGF (plasmid SH3, Supplementary Table 2) and ATG8I (plasmid SH4, Supplementary Table 2) modules were generated by recombining the pGGC0 module with the PCR products obtained on the total cDNA of *A. thaliana* using primers SH PR 7/SH PR 8 and SH PR 9/SH PR 10, respectively (Supplementary Table 1). The entry clones with ΔC versions of ATG8E and ATG8F (AM 821 and 822, Supplementary Table 2) were generated by amplifying the sequences using primer pairs AM 611/AM 643 and SH PR7/AM 643, respectively (Supplementary Table 1), entry clones with the full-length versions of the corresponding cDNAs were used as templates for the PCRs. pGGA 2x35S promoter (SH 117, Supplementary Table 2) was produced by recombining the plasmid with the PCR product using primer pair SH PR 34/SH PR 35 (Supplementary Table 1). The rest of the required modules were kindly provided by K. Schumacher's and A. Maizel's labs. The destination vectors AM 780, AM 783, AM 831, and AM 832 (Supplementary Table 2) were obtained by recombining pGGZ004, SH 117, pGGB -E2GFP-GSL, the corresponding entry clone, pGGD decoy, pGGE NosT and pGGF FAST Red pLB012. AM 784 destination clone encoding free GFP driven by the 2x35S promoter (Supplementary Table 2) was obtained by replacing the entry clones with a decoy module in the above-described recombination reaction.

SH3 and SH4 clones were also used to create a construct expressing mScarlet-tagged AtATGF and AtATG8I, respectively (SH 17 and SH19, Supplementary Table 2).

Cloning of constructs for experiments in *Chlamydomonas* was done using the modular cloning MoClo toolkit[45]. CrATG8 (Cre16.g689650, Phytozome accession number, https://phytozome-next.jgi.doe.gov/) and CrAT8 ΔC were amplified using primers ATG8_F and ATG8_R or ATG8 ΔC _R (Supplementary Table 1) and genomic DNA as a template. Amplicons were cloned into the B5 module (L0 module) of the MoClo system. Two L1 vectors were constructed using the following L0 modules: *PsaD* promoter (pCM0-016), *mCherry* gene (pCM0-067), the genomic sequence of *CrATG8* or *CrATG8* ΔC, and the *PsaD* terminator (pCM0-114). L2 vectors were assembled with hygromycin resistance. The L2 vectors were introduced into WT, *atg4*, and *atg5* strains using the glass beads methods[46].

### Establishment of Arabidopsis transgenic lines
Transgenic lines established for this study are listed in Supplementary Table 3. WT, *atg4a/b*, *atg8a-i*, and *atg5* Arabidopsis plants were transformed using the standard floral dip method[47] and *Agrobacterium*

*tumefaciens* strain GV3101. Transgenic plants were selected on the corresponding herbicide or, in the case of the FastRed selection marker, by detecting the red fluorescence in the seeds using an upright epifluorescence microscope (Axioplan, Zeiss). The fluorescent signal in the seedlings was verified using CLSM (LSM800, Zeiss and SP8, Leica). Expression of the protein with the expected molecular weight was confirmed by Western blot. The presence of the T-DNA inserts in the *atg4a/b* and *atg5* backgrounds was verified by genotyping (Supplementary Fig. 2A, B).

To obtain single *ATG4* knockout lines expressing EosFP–ATG8E ΔC or EosFP–ATG8A–CFP, crosses were performed between *atg4a/b* plants carrying the respective constructs and WT plants carrying the same constructs.

### Establishment of Chlamydomonas knockouts using CRISPR/Cas9
A CRISPR/Cas9-based targeted insertional mutagenesis approach[48] was used to knock out the *CrATG4 (Cre12.g510100*, Phytozome accession number) and *CrATG5 (Cre14.g630907*, Phytozome accession number*)* genes. For each gene, a guide RNA was designed to target the exon region near the 5' UTR (Supplementary Table 1). Electroporation was used to introduce a donor DNA comprising the *Hsp70/RBCS2* promoter, *AphVIII* gene, and *RBCS2* terminator, in addition to the ribonucleoprotein complex. The resulting transformants were screened with colony-PCR and confirmed by immunoblotting if specific antibodies were available.

### Transient expression in Arabidopsis protoplasts
For the experiment shown in Supplementary Fig. 8C, protoplasts were isolated from true leaves of 1.5-month-old plants of WT, *atg4a/b* and *atg7* backgrounds. Cells were then transformed with SH17 and SH19 constructs (Supplementary Table 2) to overexpress mScarlet–ATG8F and mScarlet–ATG8I, respectively. Protoplast isolation, transformation, and induction of autophagy were performed as previously described by Elander et al.[49] using 10 μg of the plasmid. The transfected protoplasts were let to recover in 24-well glass bottom plates (VWR CORN4441) for 16 h in light before treatment with 5 μM AZD-8055 (364424, Santa-Cruz Biotech, Dallas TX, USA) and 0.5 μM Concanamycin A (202111A, Santa-Cruz Biotech) for another 24 h. Cells were imaged using CLSM800 (Zeiss): 40×/1.2 W lens, excitation light 561 nm, and emission in the range of 570–650 nm. Images were processed using ImageJ software[50] and a dedicated macro: https://github.com/AlyonaMinina/ImageJ-macros-for-CLSM-files-procesing.

For the experiment presented in Supplementary Fig. 5 protoplasts were isolated from 4-week-old WS-0 WT and *atg4a-1/b-1* plants and transformed with the plasmids AM 779, 780, AM 831, and AM 832 (Supplementary Table 2) to express full-length and artificially truncated ATG8E and ATG8F isoforms. Cell isolation, transformation, treatment, imaging, and image analysis were performed as described above.

### Genotyping of Arabidopsis lines
Genomic DNA was extracted using a quick method. In brief, approximately 5 mg of plant material was lysed in 30 μl of 0.5 M NaOH by shaking together with glass beads, pH was then equilibrated with 370 μl of 0.1 M Tris-HCl, pH 6.8. Three μl of the resulting DNA solution was used for each PCR reaction.

*atg4a-2* mutation was verified using primer pairs annealing on the WT *ATG4A* allele (AM 409/AM 587, Supplementary Table) and on the *atg4a-2* T-DNA insertion (AM 410/AM 584, Supplementary Table 1). *atg4b-2* mutation was verified using primer pairs annealing on the wild-type *ATG4B* allele (AM 423/AM 420, Supplementary Table 1) and on the *atg4b-2* T-DNA insertion (AM 420/AM 584, Supplementary Table 1). Detection of the *atg5-1* T-DNA insertion was done using primers AM

1/AM 579 (Supplementary Table 1), while the WT ATG5 gene was detected with primers AM1/AM2 (Supplementary Table 1).

### Isolation of the Chlamydomonas genomic DNA
Genomic DNA was isolated as previously described[51]. In brief, 10 ml of cells at the late log phase were harvested and resuspended in 150 μl H$_2$O and subsequently in 300 μl of SDS-EB buffer (2% [*w/v*] SDS, 400 mM NaCl, 40 mM EDTA, 100 mM Tris-HCl, pH 8.0). The mixture was mixed with 350 μl phenol: chloroform:isoamyl alcohol (25:24:1, *v/v/v*) by inversion several times, and two phases formed after centrifugation at 2000× *g* for 5 min at room temperature. The upper aqueous phase (300 μl) was collected and extracted using 300 μl chloroform:isoamyl alcohol (24:1, *v/v*). The resulting aqueous phase was blended with 2 volumes of absolute ethanol and kept at −80 °C for 30 min. The genomic DNA was collected by centrifugation at 17,000× *g* for 10 min at room temperature and washed with 200 μl 70% (*v/v*) ethanol twice. The genomic DNA was air-dried and dissolved in 200 μl H$_2$O.

### RT-PCR
Total RNA was extracted from 1.5-week-old Arabidopsis seedlings pooling 3 seedlings/extraction for each transgenic line and using Spectrum™ Plant Total RNA Kit (Sigma-Aldrich, STRN250), RNAs were on-column treated with the DNase (Sigma-Aldrich, DNASE70-1SET) following the kit instructions. One μg of total RNA was used for each RT reaction implementing the Maxima First Strand cDNA Synthesis Kit (ThermoFisher, K1641) and oligodT primers provided with the kit, and following the manufacturer's protocol. Full-length cDNAs of the ATG4A, ATG4B, and WT were detected using primer pairs AM 417/AM 418, AM 419/AM 420, and AM 8/AM 9, respectively (Supplementary Table 1). The amplicons were separated in a 1% TAE agarose gel containing GelRed dye and imaged with ChemiDoc (Bio-Rad).

### qPCR
Primers for ATG8 isoforms were designed and checked for specificity against Arabidopsis transcriptome using Primer-Blast (Supplementary Table 1). Primers for reference genes were taken from the previous study[52]. Total RNA extraction and RT were performed as described above, at least 15 seedlings were pooled for each line for the RNA extraction. RNA RIN was confirmed to be above 8. One thousand ng of total RNA were used for each cDNA synthesis reaction, RT reactions were diluted 1:25, and 5 μl were used for each qPCR reaction (final volume of 15 μl), and each reaction was run in triplicates. qPCR was performed using the DyNAmo Flash SYBR Green qPCR Kit (Thermo-Fisher, F-415L), the recommended in the manual two-step protocol supplemented with the melt curve step. Primers were used at the 0.5 μM final concentration and annealed at 60 °C. The assay was carried out in the CFX Connect (Bio-Rad) and analyzed using CFX Maestro software (Bio-Rad). The lack of off-target amplification was verified by the melt curve analysis and stability of the reference genes was confirmed by ideal *M*-values identified by the software. Expression of each GOI was normalized to all three reference genes using the ΔΔCt method and presented as the fold change relative to the WT sample devoid of any transgenes.

### In planta ATG8-CFP cleavage assay
True leaves of one-month-old Arabidopsis plants expressing EosFP–ATG8A–CFP in *atg4a, atg4b, atg4a/b,* and WT backgrounds were harvested and immediately flash-frozen in liquid nitrogen. Plant material was powdered while frozen and mixed 1:2 (*m:v*) with a pre-heated 2× Laemmli buffer. Samples were boiled for 10 min at 95 °C and centrifuged at 17,000×*g* for 10 min. Seven and a half μl of each supernatant were loaded on a 4–20% TGX Stain-Free precast gel (Bio-Rad, 4568096). Separated proteins were transferred onto the PVDF

membrane (Bio-Rad, 1620174) and blotted with anti-GFP (Roche, 11814460001) and anti-EosFP (Encorbio, RPCA-EosFP), both at 1:2000 dilution. Anti-mouse (Amersham, NA931-1ML) and anti-rabbit (Invitrogen, a11008) HRP conjugates were used at 1:5000 and 1:20,000 dilutions, respectively. The signal was developed using SuperSignal™ West Femto Maximum Sensitivity Substrate (ThermoFisher, 34094) and Chemi-Doc Imaging System (Bio-Rad). Densitometric analysis was performed as described in the dedicated chapter below.

### In vitro ATG8-CFP cleavage assay

AM810–813 plasmids (Supplementary Table 2) encoding active and proteolytically dead (PD) GST-tagged AtATG4A and AtATG4B were used to transform Rosetta (DE3) cells (Novagen, 70954-3). Expression was induced overnight at 16 °C using 0.25 mM IPTG (Hellobio, HB3941). Cell cultures were harvested by centrifugation at 5000×$g$ for 20 min at 4 °C, and the cell pellets were stored at −80 °C until purification. Subsequent steps were conducted at 4 °C. Cells were resuspended using a ratio of 1 g to 5 ml of binding buffer containing 20 mM PBS buffer, and 100 mM NaCl, at pH 7.3. Cells were lysed using a cell disruptor at 20 kpsi. The resulting sample was then treated with DNase I for 30 min. Insoluble cell debris was eliminated through centrifugation at 38,000 $g$ for 20 min, followed by filtration of the supernatant through a 0.45 μm filter. The filtered supernatant was applied to an 8.5 mm diameter column packed with 1 ml of Glutathione Sepharose 4B matrix (Cytiva, 17075601) and equilibrated with the binding buffer. After sample application, the column was washed with approximately 10 column volumes of the binding buffer. The protein was eluted using an elution buffer containing 50 mM Tris-HCl, 100 mM NaCl, and 10 mM reduced glutathione, at pH 8.0. Each fraction was collected after a 5-min incubation with the elution buffer. The protein purity was assessed on 4–20% gradient TGX SDS-PAGE gels. The purest fractions were combined and concentrated to 1 ml using Vivaspin 30 K centrifugal concentrators at 4 °C (Sartorius, VS0621). The protein sample was then further purified by size exclusion chromatography on a HiLoad 16/600 Superdex 200 pg column (Cytiva, 28989335) connected to an ÄKTA FPLC system (Cytiva). The column was equilibrated with a buffer containing 50 mM Tris-HCl, 100 mM NaCl, and 1 mM DTT at pH 8.0. Protein fractions were analyzed with SDS-PAGE, pooled, and concentrated to the desired concentration. The protein concentration was calculated based on the measured A280 nm using a NanoDrop 1000 Spectrophotometer and the theoretical molar extinction coefficient was determined using the ProtParam tool on the ExPASy server[52]. Before storing at −80 °C, the protein sample was mixed with glycerol in a 1:1 ratio.

True leaves of one-month-old Arabidopsis plants expressing EosFP–ATG8A-CFP in *atg4a/b* and WT backgrounds were harvested and immediately flash-frozen in liquid nitrogen. Plant material was powdered while frozen and mixed 1:2 (*m:v*) with reaction buffer containing 50 mM Tris pH 7.5, 100 mM NaCl, 5 mM DTT, 5 mM EDTA, 5 mM PMSF, 0.05% Tween-20, and 50 ng of purified active or PD ATG4. Reactions were kept at room temperature for the indicated period of time, prior to being mixed 1:1 with a preheated 2× Laemmli buffer. Detections of EosFP and CFP in the reaction were performed using Western blot as described for the *in planta* ATG8-CFP cleavage assay. Densitometric analysis was performed as described in the dedicated chapter below.

### In vitro ATG8 delipidation assay

True leaves of one-month-old Arabidopsis plants expressing EosFP–ATG8E ΔC in *atg4a/b* and WT backgrounds were harvested and immediately flash-frozen in liquid nitrogen. Plant material was powdered while frozen and mixed 1:2 (*m:v*) with reaction buffer containing 50 mM Tris pH 7.5, 100 mM NaCl, 5 mM DTT, 5 mM EDTA, 5 mM PMSF, 0.05% Tween-20, and 50 ng of purified active or PD GST-ATG4. Reactions were kept at room temperature for 30 min, prior to being mixed

1:1 with a preheated 2× Laemmli buffer. To estimate the original amount of ATG8 and ATG8-PE present in the samples, quick protein extraction was performed by mixing an aliquot of frozen plant material directly with a preheated 2× Laemmli buffer at 1:1 *m:v* ratio. Proteins from the quick extracts and delipidation reactions were separated on the manually cast 10% PAAG supplemented with 6 M urea[23,53]. To ensure sufficient separation of the tagged ATG8 adducts, the electrophoresis was performed until the 25 kDa band of the prestained PageRuler (ThermoFisher Scientific, 26619) reached the bottom edge of the gel. Separated proteins were transferred onto the PVDF membrane (Bio-Rad, 1620174). Detection of EosFP was performed as described for the *in planta* ATG8-CFP cleavage assay. Densitometric analysis was performed as described in the dedicated chapter below.

### Autophagy induction

Induction of autophagy in 7-day-old Arabidopsis seedlings was performed in liquid 0.5× MS medium complemented with 5 μM AZD8055 (Santa-Cruz Biotech, 364424) and 0.5 μM concanamycin A (Santa-Cruz Biotech, 202111A). As verified by the accumulation of autophagic bodies and by GFP-cleavage assay, treatment for 2 h was sufficient for induction of autophagy in seedlings roots, while treatment for 24 h was used for cotyledons.

For autophagy induction in older plants, the abaxial side of true leaves was infiltrated with MQ water containing 5 μM AZD and 0.5 μM ConA. The adaxial side of the infiltrated leaves was imaged after 24 h of treatment.

Treatments with 10 mM N-ethylmaleimide (NEM, Sigma-Aldrich 04259) and 1 mM iodoacetamide (IAM, Cytiva RPN6302) were performed simultaneously with AZD and ConA due to high phytotoxicity of the NEM and IAM. These treatments were performed using the approach described above for AZD/ConA. E64d showed lower toxicity in Arabidopsis, therefore plant material was pretreated with 5 μM E64d (Sigma-Aldrich E8640) prior to autophagy induction with AZD/ConA.

Nitrogen-depleted conditions were implemented by growing the seedlings starting from seed germination on 0.5× MS devoid of nitrogen source: 0.5× MS basal salt mix (Sigma-Aldrich M0529): 1.5 mM CaCl₂, 0.75 mM MgSO₄, 0.25 mM KH₂PO₄, 2.5 mM KCl, 1% sucrose, and 0.8% Plant agar.

Carbon-depleted conditions were achieved by using 0.5× MS (described above) without sucrose and keeping plants in the dark.

### Confocal microscopy

Confocal microscopy was performed using CSLM SP8 (Leica) and CLSM 800 (Zeiss), in either case using a 40× water immersion lens, NA = 10–1.2. GFP, CFP, and EosFP were imaged using an excitation light of 488 nm and an emission range of 515–560 nm. Red fluorescence was detected using 561 nm excitation light and 570–650 nm emission.

To prevent root desiccation during imaging, seedlings were mounted inside RoPod chambers[55], in a droplet of liquid corresponding to the treatment. Detached cotyledons were imaged while mounted between two coverslips. Ten by ten mm sections were cut out of true leaves immediately prior to imaging and mounted in a droplet of liquid between a standard microscopy slide and a coverslip additionally secured with tape. True leaves and cotyledons were imaged only from the adaxial side.

Confocal images were processed using LAS X (for Leica micrographs) software or ImageJ[50] implementing the dedicated macro for channel separation and image cropping: https://github.com/AlyonaMinina/ImageJ-macros-for-CLSM-files-procesing.

### Quantification of autophagic bodies density and area

Quantification of autophagic bodies was performed in a semi-automated manner using ImageJ software and custom-made macro. The puncta quantification shown in Fig. 2, Supplementary Figs. 3 and 4 was done using the macro for puncta quantification in single channel

v4. It can be found in the dedicated GitHub repository: https://github.com/AlyonaMinina/Puncta-quantification-with-ImageJ.

The quantifications shown in Supplementary Figs. 4 and 9 were obtained using a macro for autophagic body size and density measurement: https://github.com/AlyonaMinina/Autophagic-bodies-size-measurement.

Statistical analysis of the quantitative data obtained with ImageJ macros was performed using custom R scripts. Unless stated otherwise, results were subjected to Tukey's HSD test. Plots were built using the ggplot2 R package[56]. Each experiment was performed at least twice, using at least three biological replicates (individual plants), three technical replicates for imaging (images of representative areas on the same plant), and five to ten technical replicates for measurements (regions of interest selected on each image using the corresponding macro script).

### TEM
Seven-day-old seedlings were treated with 5 $\mu$M AZD and 0.5 $\mu$M ConA, in 0.5× MS liquid medium, subjected to confocal microscopy, and then fixed in the solution containing 2.5% glutaraldehyde and 1% paraformaldehyde in 0.1 M phosphate buffer. Further sample processing was performed at BioVis, Uppsala: the samples were post-fixed with 1% Osmium tetroxide, dehydrated in ethanol embedded in resin. Ultrathin sections were collected on copper slot grids and contrasted with uranyl acetate and lead citrate. Sections were imaged using FEI Technai G2 operated at 80 kV. Images were processed using ImageJ 1.53c.

### Western blotting
Whole Arabidopsis seedlings or a section of true leaves were sampled into liquid nitrogen and powdered while frozen. A pool of 4−8 seedlings per sample or sections from three different leaves was used as one sample to ensure the representativity of the data. To limit potential protein proteolysis during sample preparation, the powdered material was directly mixed with preheated Laemmli buffer in a 1:2 (m:$v$) ratio. Samples were boiled at 95 °C for 10 min, mixing every 2 min. Debris was removed by centrifugation at 17,000 $g$ for 10 min at RT. Supernatants were transferred into new Eppendorf tubes and stored at −20 °C. Typically, 7.5–15 $\mu$l of each sample was used for separation on the gel. Concentrations of all antibody stock solutions were according to the recommendations of the Ab providers.

To separate tagged ATG8 and tagged ATG8−PE adducts, 10% PAAG was prepared manually supplementing it with 6 M urea[23,54]. To ensure sufficient separation of the tagged ATG8 adducts, the electrophoresis was performed until the 25 kDa band of the prestained PageRuler (ThermoFisher Scientific, 26619) reached the bottom edge of the gel. For other Western blot assays, proteins were separated on 4−20% TGX Stain-Free precast gels (Bio-Rad, 4568096). Detections of the EosFP, GFP, and CFP were performed using Western blot assay as described for the *in planta* ATG8−CFP cleavage assay.

Total proteins from *Chlamydomonas* cells were extracted as described in ref. 51. In brief, 1 ml of cell cultures in the late log phase (1−2 × 10$^7$ cells ml$^{-1}$) were harvested for 1 min centrifugation at 17,000× $g$ and were snapfrozen in liquid nitrogen. The cell pellet was dissolved in 100 $\mu$l 1× SDS sample buffer (50 mM Tris-HCl pH 7.5, 100 mM DTT, 2% [$w/v$] SDS, 0.1% [$w/v$] bromophenol blue, 10% [$w/v$] glycerol) and incubated at 80 °C for 3 min. The mixture was vortexed at the highest speed and then spun down at 17,000× $g$ for 10 min at room temperature. The resulting supernatant was collected for immunoblotting. Anti-CrATG4 (Agrisera, AS15 2831) and Anti-CrATG8 (Agrisera, AS14 2769) were used at a dilution of 1:5000, while anti-rabbit HRP-conjugate (Agrisera, AS07 260) at a dilution of 1:10,000.

### Densitometric analysis
Densitometric analysis of the western blots was performed using a custom-designed ImageJ macro. GFP-cleavage assay macro was used for EosFP/GFP-cleavage assay shown in Figs. 2E, 3C, and 5D and

Supplementary Figs. 2F, 2G, 4D, and 9D. The script and instructions for users are available on GitHub: https://github.com/AlyonaMinina/NACA-N-terminal-ATG8-cleavage-assay.

Densitometric analysis for the C-terminal ATG8 cleavage assay shown in the Supplementary Fig. 2F and G, and Supplementary Fig. 3D was carried out using the dedicated ImageJ macro (https://github.com/AlyonaMinina/CACA-C-terminal-ATG8-cleavage-assay).

For Fig. 2D, Supplementary Figs. 3G, 3K, 3L, and 4E ATG8-lipidation was quantified using the ALDA macro version for western blots with two bands; while quantification for Fig. 5E was made using the macro version for western blots with three bands. Both ImageJ macros are available on GitHub: https://github.com/AlyonaMinina/ALDA-ATG8-lipidation-densitometric-assay.

Quantitative ImageJ macro outputs were further processed in R Studio using custom R scripts for sample relabeling and Tukey's HSD test, plots were built using the ggplot2 R package[56].

### Plant phenotyping
Root growth assays were performed using SPIRO[24] for automated imaging of Petri plates and SPIRO assays for image analysis. To ensure the reproducibility of the conditions and therefore comparability of the results, SPIRO robots were placed inside the same plant growth cabinets that were used for growing seedlings on plates in other experiments of this study. For experiments illustrated in Fig. 4, Supplementary Movies 2 and 3, and Supplementary Fig. 6 seeds were plated directly on the corresponding medium used for the stress induction, and plates were imaged without interruption during day and night for at least 7 days, utilizing the green LED of SPIRO for imaging in the dark. For carbon starvation assays, seedlings were allowed to grow under long-day conditions for 4−5 days, followed by 4 days of dark treatment and another 4 days of recovery. SPIRO data analysis was performed as described by Ohlsson et al. in ref. 24.

For the experiments shown in Fig. 3D, E, Supplementary Fig. 4F, G, and Figs. 5I and 4J homozygous lines with no somatic silencing were not available, therefore we made sure to first confirm a good level of transgene expression prior to phenotyping. For this, seeds of the T1 and T2 generation, showing the red fluorescent signal of the FastRed marker, were first germinated on the standard 0.5× MS medium. Roots of 4- or 5-day-old seedlings were checked with a stereoscope to ensure a strong and even GFP signal. Seedlings with confirmed transgene expression were then transferred to control or −N medium and imaged using SPIRO for up to 14 days. For the analysis, seedlings of the T2 generation line were treated as a single individual line, while seedlings from the T1 generation were pulled together to form another "individual line" comprising a pool of unique transformation events. For quantifying root growth, complete root systems were traced using the SNT ImageJ toolbox[57].

Photos of plants in soil were obtained using the camera of an iPhone XI and further processed in Photoshop 25.0.0.

### Senescence assay in Chlamydomonas
The *Chlamydomonas* cells were pre-cultured until they reached the early stationary phase. The cells were then subjected to ten times dilution using fresh TAP medium. Upon attaining the late log phase or stationary phase, 1 × 10$^5$ cells were inoculated on TAP agar plates for senescence assay. Cell colonies were imaged after 40 days.

### Protein sequence alignment and phylogenetic tree building
Protein sequence alignments and construction of the phylogenetic tree were done using the Geneious software v10.2.6.

### Statistics and reproducibility
Each experiment was repeated at least twice, and no experiments were excluded from the data analysis due to biological reasons. Statistical analysis was performed on biological replicates ($n$ value indicated in

figure legends), with data represented by multiple technical replicates. CI for the statistical tests was set to 95%. No preselection of samples or groups was done in any experiment; grouping was based on the biological question (e.g., treatment, genotype). Boxes in the box plots represent the interquartile range (IQR), with the median indicated by a horizontal line. Whiskers extend to 1.5 times the IQR, and outliers are shown as filled circles. Error bars on the bar charts represent the standard error (SE).

SPIRO root growth assay results show data obtained by fitting a mixed-effect second-order polynomial model to the time-resolved root length data obtained by the SPIRO root growth assay[24].

For the sake of reproducibility, we provide scripts for the semi-automated image analysis. Raw data and intermediate analysis steps recorded by the scripts will be provided upon request.

### Reporting summary
Further information on research design is available in the Nature Portfolio Reporting Summary linked to this article.

## Data availability
Source data are provided in this paper. Raw CLSM data and SPIRO time-lapse imaging data will be provided by the corresponding author upon request. Source data are provided in this paper.

## Code availability
Image analysis was done using ImageJ macros as referenced above. For reproducibility, the commit hashes corresponding to the source code at the time of analysis are given within brackets for each macro: https://github.com/AlyonaMinina/ImageJ-macros-for-CLSM-files-procesing (https://doi.org/10.5281/zenodo.14508717); https://github.com/AlyonaMinina/Puncta-quantification-with-ImageJ (https://doi.org/10.5281/zenodo.14506480); https://github.com/AlyonaMinina/Autophagic-bodies-size-measurement (https://doi.org/10.5281/zenodo.14506416); https://github.com/AlyonaMinina/NACA-N-terminal-ATG8-cleavage-assay (https://doi.org/10.5281/zenodo.14508441); https://github.com/AlyonaMinina/CACA-C-terminal-ATG8-cleavage-assay (https://doi.org/10.5281/zenodo.14508708); https://github.com/AlyonaMinina/ALDA-ATG8-lipidation-densitometric-assay (https://doi.org/10.5281/zenodo.14509069). Germination assessment and root growth analysis were done using the SPIRO Germination and Root Growth assay: https://github.com/jiaxuanleong/SPIRO.Assays (https://doi.org/10.5281/zenodo.14509658).

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

## Acknowledgements

We would like to thank Monika Hodik and Karin Staxäng at the EM facility, BioVis, Uppsala for help with the TEM experiments. This study was supported by grants from EU Horizon 2020 MSCA IF (799433) and Carl Tryggers Foundation (CTS 14 326 and 20:287) to E.A. Minina; the Swedish Research Council Formas (2017-00541 to P.V. Bozhkov and 2021-01812 to E.A. Minina); the Swedish Research Council Vetenskapsrådet (2019-04250 to P.V. Bozhkov, E.A. Minina, and P.N. Moschou); the Knut and Alice Wallenberg Foundation (2018.0026 to P.V. Bozhkov and P.N. Moschou; 2021.0071 to S. Stael), ERC (101044878 to S. Stael and 101126019 to P.N.M.), SFB1101 (TPA02, DFG to K. Schumacher); Austrian Science Fund (FWF-P34944 to Y. Dagdas); European Research Council Grant (101043370 to Y. Dagdas and 948996 to S. Uestuen); Emmy Noether Fellowship GZ: UE 188/2 (DFG to S. Uestuen) and crops for the future research program at the Swedish University of Agricultural Sciences (to P.V. Bozhkov and to E.A. Minina). Views and opinions expressed are those of the author(s) only and do not necessarily reflect those of the European Union or the European Research Council Executive Agency. Neither the European Union nor the granting authority can be held responsible for them.

## Author contributions

Y.Z. performed all experiments, figure preparation, and manuscript writing involving *Chlamydomonas*; J.A.O. contributed to the analysis of experiments and editing of figures and the manuscript; S.H. contributed to the analysis of experiments and editing of figures; I.S. contributed to the analysis of experiments and manuscript editing; J.X.L. contributed analysis of experiments and manuscript editing; F.B. contributed analysis of experiments and manuscript editing; M.K., K.S., S.U., and Y.D. provided unpublished material and contributed to manuscript editing; P.N.M, S.S, and P.V.B. contributed editing of figures and the manuscript; and E.A.M. contributed the study concept; planning, performance, and analysis of experiments; figure preparation and writing of the manuscript.

## Funding

## Competing interests

The authors declare no competing interests.
