## [Peer review File · Nature Communications]

Reviewers' comments:

Reviewer #1 (Remarks to the Author):

This work by Holla et al. proposed an intriguing hypothesis that the process of ATG8 delipidation is not essential for autophagy in plants. They found ATG8-C/ATG8I labelled autophagosome-like structures was found in the vacuoles of the *atg4a/b* mutant, leading to the conclusion that ATG4-dependent ATG8 delipidation is dispensable for maintaining autophagic flux. Unfortunately, most of the conclusions are rather preliminary, and in many aspects inconsistent.

Major concerns:

1. I think the experimental data provided in the manuscript do not sufficiently support the main conclusions, particularly due to an excessive reliance on the overexpression system for functional studies.

For the entire Figure 3, only one transgenic line was analysed in each genetic background, and their expression of ATG8E -C is not known. The same issue is true for Figure 4 as well, it is important to use multiple lines for phenotype studies, as variations are expected when using over-expression lines.

2. In many places (e.g. Figure 1D, Figure 4F), the expression level of ATG8 was not given, I believe more autophagosomes are expected with higher expression of these ATG8 variants. Did the author consider the potential variations produced by protein expression level?

3. Overexpressing an artificially constructed ATG8E Δ C for investigating the impact of ATG8 delipidation is not a suitable method. Could these phenotypes in Figure 3 (assuming they are repeatable in independent transgenic lines) be an artefact?

I know it is technically challenging, but the most reliable method to analyze the function of ATG8E Δ C and ATG8I is to use an ATG8 loss-of-function mutant. Otherwise, we cannot say ATG8E Δ C is functional, or ATG8 lipidation is dispensable.

4. Only light microscopy was used for autophagosome analysis is not sufficient here, the authors should consider using TEM to confirm some of the results in Figure 1D, and Figure 4F. Are these ATG8-C or ATG8I labelled puncta in *atg4* mutant real autophagosomes? Ultrastructural studies are required to address this issue.

5. If ATG4 was the only protease acting on the C-terminal peptide of ATG8 protein, then the EOSfP-ATG8-CFP should not generate free CFP in atg4 mutant; however, this is not the case in Figure S2D-E. This also indicates that other unknown proteins that functionally homologue to ATG4 may exist.

6. No further information is given about the atg4a/atg4b double mutant, is it a true KO? According to the material and methods, this mutant line was adopted from a previous study: atg4a-2 (SAIL_740_H03) and atg4b-2 (SALK_056994). However, the T-DNA is inserted at the very end of the atg4b-2 coding sequence, in this case, most of the ATG4b gene should still be expressed.

If the atg4a/atg4b is a partially functional mutant, it may explain major point 5 nicely. In any way, further confirmation is needed.

6. The quantification of autophagosomes and the western blot analysis for the level of free-EosFP were essential in Figure 2C.

7. Some quantifications are required for the immune-blot, such as Figure 2D-E, Figure 4D-E, Figure S2D-E, and Figure S4C-D.

8. Figure 5, the failure of the truncated ATG8 version to rescue the early senescence phenotype in the cratg4 mutant may be attributed to deficiencies in autophagy rather than abnormal ATG8 delipidation. Firstly, it is necessary to verify whether autophagy flux is restored in cratg4 by complementing it with mCherry-tagged full-length CrATG8.

9. If delipidation is not necessary, we should not find the autophagy-related phenotype in the atg4a/atg4b mutant, as ATG8H and ATG8I should still be functional. Here, the author further suggested that "...the sum amount of ATG8H and ATG8I is much lower than the sum amount of all isoforms ATG8A-ATG8I available for lipidation in the WT background and might not be sufficient for normal autophagosome biogenesis", or "ATG8H and ATG8I isoforms play specific roles in plant autophagy that do not include participation in autophagosome biogenesis". However, these hypotheses are not sustained and require further confirmation.

Reviewer #2 (Remarks to the Author):

Autophagy is a process conserved in eukaryotic cells. Yet there are specific regulations in each species. In this study, Holla et al. reported an interesting divergence in plant autophagy: vascular plant *Arabidopsis* does not necessarily require an ATG4-dependent ATG8 delipidation in stress-induced autophagy, but *Chlamydomonas* does. This discovery makes a step forward in our understanding of the maturation of autophagosomes in plant responses to stresses. The data generated using a combined approach of genetics, biochemistry and imaging are solid, and all experiments were well designed. There are several minor concerns:

(1) Line 82-84: The manuscript states that ATG8E Δ C cannot be delipidated in the absence of ATG4. It is too early to state this without supporting data. It would be beneficial for the authors to modify this and let the readers to get the point after reading the related data (e.g. Fig. S4) presented later on.

(2) Majority of Western blot images presented are without size markers.

(3) A positive control demonstrating the effectiveness of NEM or IAM should be included In Figure 2C to validate the experimental results.

(4) The statement in line 161-162 ‘... normally-functioning autophagosome formation and delivery to the vacuole under these conditions.’ may be bit too strong. Figure 2B indicates that there is a decrease in ATG8e Δ C delivery to the vacuole in *atg4/b* mutants, at least in shoots?

(5) I suggest changing the title to ‘ATG8 delipidation is dispensable for autophagy in *Arabidopsis*’, as there were no plants other than *Arabidopsis* tested.

(6) Numerous references in the manuscript are incorrectly formatted and require correction.

We would like to express our gratitude to the reviewers for their valuable feedback. In essence, we addressed all the comments raised by the reviewers and added a substantial amount of new data including:

- (i) Confirmation that, akin to the ATG8E Δ C isoform, an artificially primed ATG8F isoform can effectively restore autophagic activity in the ATG4 loss-of-function mutant (**Fig. 2I** and **Fig. S4**).
- (ii) Validation of our main finding using an alternative ATG4 loss-of-function mutant (**Fig 2J** and **Fig. S6**).
- (iii) Ultrastructural TEM data, providing additional support for the results obtained using CLSM, plant phenotyping, and biochemistry assays (**Fig. 5**).
- (iv) Further control experiments and quantifications suggested by the reviewers (**Fig. 2D** and **E**, **Fig. 4D** and **E**, **Fig. S2D, F** and **G**; **Fig. S3C, D, I-L**; **Fig. S5, Fig. S8D, Fig. S9D**)

We believe that revised manuscript makes a much stronger claim now and are grateful to the reviewers for their constructive comments and suggestions. Please find our point-by-point replies to the reviewers' comments in the text below.

Reviewer #1

Comment 1.1: I think the experimental data provided in the manuscript do not sufficiently support the main conclusions, particularly due to an excessive reliance on the overexpression system for functional studies.

For the entire Figure 3, only one transgenic line was analysed in each genetic background, and their expression of ATG8E -C is not known. The same issue is true for Figure 4 as well, it is important to use multiple lines for phenotype studies, as variations are expected when using over-expression lines.

Answer 1.1: We agree with the reviewer that phenotyping a single line is insufficient. Unfortunately, the number of T-DNA insertions in the genome of *atg4a/b* plants renders them especially susceptible to somatic silencing of additional transgenes (**Fig. S1**). Despite our efforts, we were unable to identify more lines expressing ATG8E Δ C with the stability suitable for phenotyping. We addressed this limitation by performing another round of plant transformation and seized the opportunity to investigate whether the effect observed with ATG8E Δ C could be reproduced with another artificially primed ATG8 isoform, viz. ATG8F Δ C.

In the revised version of the manuscript, we demonstrate that expression of ATG8F Δ C can also restore autophagic activity in the ATG4-deficient plants. We have checked at least two independent lines for each genotype in each experiment involving these lines. In total more than 20 individual lines for WT and *atg4a/b* were used.

Please note that **Figure 4** contains examples of 2-4 independent lines per genotype (except for WT and KOs not expressing tagged ATG8s). We have further clarified this information in the figure legend. Concerning the transgene expression level and detection of the expressed proteins please refer to the reply to comment 1.2.

Comment 1.2: In many places (e.g. Figure 1D, Figure 4F), the expression level of ATG8 was not given, I believe more autophagosomes are expected with higher expression of these ATG8 variants. Did the author consider the potential variations produced by protein expression level?

Reply 1.2: In the revised version we provide qPCR data showing expression of full-length and artificially primed ATG8E (**Fig. S2D**) and ATG8F (**Fig. S4C**) isoforms, as well as natively truncated ATG8I isoform (**Fig. S8D**) in the transgenic lines used for this study.

Please note, that expression of ATG8E Δ C in WT and in *atg4a/b* seedlings and true leaves of soil-grown plants is shown on several western blots (WBs) (**Fig. 2D** and **E**, **Fig. S3K** and **L**). Expression levels of ATG8F and ATG8I in WT and *atg4a/b* are shown on the WB in **Fig. 4D** and **E**, as well as **Fig. S9D**. Expression of ATG8F Δ C is also corroborated by WB analysis (**Fig. 4D** and **E**).

Our results demonstrate that even a very strong overexpression of a full-length ATG8 does not allow for the formation of autophagic bodies in *atg4a/b* plants, confirming the necessity of ATG8 priming by the ATG4 protease for this process.

Comment 1.3: Overexpressing an artificially constructed ATG8E Δ C for investigating the impact of ATG8 delipidation is not a suitable method. Could these phenotypes in Figure 3 (assuming they are repeatable in independent transgenic lines) be an artefact?

I know it is technically challenging, but the most reliable method to analyze the function of ATG8E Δ C and ATG8I is to use an ATG8 loss-of-function mutant. Otherwise, we cannot say ATG8E Δ C is functional, or ATG8 lipidation is dispensable.

Reply 1.3: We have added more experiments to demonstrate that autophagic activity can be observed in the *atg4a/b* upon overexpression of three different versions of ATG8: artificially primed ATG8E Δ C (**Fig. 2A, B, E**,

J) or ATG8F Δ C (Fig. 2C and J, Fig. S4), and natively primed ATG8I (Fig. 4). Therefore, the observed autophagic activity is not an artefact of ATG8E Δ C expression.

Furthermore, we confirmed functionality of the artificially primed ATG8E Δ C and ATG8F Δ C in the ATG8 nonuple loss-of-function mutant. For these experiments the nonuple knockout of all nine Arabidopsis ATG8 genes was kindly provided by Y. Dagdas group. Since the original publication describing this knockout is still under preparation, we are showing the results requested by the reviewer only in the rebuttal letter. However, we would like to incorporate these results into our study if the work describing the knockout is released before our study is accepted for publication:

“To eliminate the possibility of unforeseen artifacts associated with using artificially primed ATG8s, we conducted an additional control experiment validating the functionality of both ATG8E Δ C and ATG8F Δ C. For this, the nonuple knockout of all nine Arabidopsis ATG8 genes was transformed with the constructs for expression of the corresponding GFP-tagged ATG8 isoforms or free GFP. The transformation and transgenic lines selection was carried out as described in the materials and methods section for other Arabidopsis backgrounds. Transgenic 7-day-old seedlings were subjected to AZD/ConA treatment followed by confocal microscopy detecting accumulation of autophagic bodies in the vacuoles. Either variant of the ATG8 was capable of restoring autophagic activity in the ATG8 loss-of-function mutant (Figure).”

Either of the artificially primed ATG8F or ATG8E isoforms is sufficient to restore autophagic activity in the absence of other ATG8 isoforms.

A. Autophagy-deficient phenotype of the nontuple ATG8-knockout line was verified using SPIRO assay to track root growth on control and nitrogen-depleted media. Representative SPIRO image of 5-day-old seedlings on both types of media. Scale bar, 1 cm.

B. Results of the SPIRO Root Growth assay for samples illustrated in (A). The chart shows predicted root lengths at 96 h after seed germination. Error bars represent SE. Distinct letters represent groups that are significantly different from each other, Tukey's HSD test, $\alpha = 0.05$. The chart shows representative results of one out of two experiments, $n = 218$.

C. Confocal microscopy images of root epidermal cells. Prior to imaging, 7-day-old Arabidopsis seedlings expressing GFP-ATG8E Δ C, GFP-ATG8F Δ C or GFP in nontuple ATG8-knockout were treated for 2 h with AZD/ConA to induce autophagy or were kept under normal conditions (mock). Four independent lines for each construct were checked to verify reproducibility of the results. White arrowheads point at autophagic bodies accumulating in the vacuoles. No autophagic bodies were observed in the vacuoles of seedlings expressing only GFP. Scale bars, 20 μ m.

Comment 1.4: Only light microscopy was used for autophagosome analysis is not sufficient here, the authors should consider using TEM to confirm some of the results in Figure 1D, and Figure 4F. Are these ATG8-C or

ATG8I labelled puncta in *atg4* mutant real autophagosomes? Ultrastructural studies are required to address this issue.

Reply 1.4: We agree with the reviewer and performed the suggested TEM experiment, which indeed revealed autophagic bodies in the vacuoles of *atg4a/b* plants expressing primed versions of ATG8 (**Fig. 5**).

Comment 1.5: If ATG4 was the only protease acting on the C-terminal peptide of ATG8 protein, then the EosFP-ATG8-CFP should not generate free CFP in *atg4* mutant; however, this is not the case in Figure S2D-E. This also indicates that other unknown proteins that functionally homologue to ATG4 may exist.

Reply 1.5: We agree with the reviewer that a free CFP band in the *atg4a/b* KO protein extract indicates proteolysis of the EosFP-ATG8-CFP fusion by a protease different from ATG4. It is worth noting that the CFP band released in the *atg4a/b* plants has a smaller MW (compared to the CFP released in the WT background) indicative of a possible cleavage *within* the CFP protein. Critically, such proteolysis does not lead to accumulation of the EosFP-ATG8 adduct required for autophagy. Furthermore, the ATG8-CFP product is detectable only in the *atg4a/b* KO protein extracts, indicating that cleavage of CFP from the C-terminus of ATG8 is severely impaired in the absence of ATG4. To aid interpretation of this Western blot we provided results of densitometry analysis in **Fig. S2F**.

Comment 1.6: No further information is given about the *atg4a/atg4b* double mutant, is it a true KO? According to the material and methods, this mutant line was adopted from a previous study: *atg4a-2* (SAIL_740_H03) and *atg4b-2* (SALK_056994). However, the T-DNA is inserted at the very end of the *atg4b-2* coding sequence, in this case, most of the ATG4b gene should still be expressed.

If the *atg4a/atg4b* is a partially functional mutant, it may explain major point 5 nicely. In any way, further confirmation is needed.

Reply 1.6: We agree with the reviewer that this is a critical point for our study and even dedicated an entire supplementary **Figure S2** to this point. To reiterate our conclusions: we consider *atg4a-2/b-2* (further *atg4a/b*) a true loss-of-function ATG4 mutant based on several lines of evidence.

First, we extensively verified the *atg4a/b* plants through genotyping (**Fig. S2A and B**), RT-PCR (**Fig. S2C**) and the ATG4 activity assays performed *in vivo* (**Fig. S2E and F**) and *in vitro* (**Fig. S2G**). These analyses confirmed the presence of the expected T-DNA insertions in both alleles of the genes of interest, the absence of the full-length mRNA for both ATG4A and ATG4B and the lack of ATG4-mediated processing necessary for generating autophagy-relevant ATG8 adducts.

Second, no autophagic bodies are detectable in the *atg4a/b* plants expressing an ATG8 containing the C-terminal peptide (ATG8E full length, **Fig. 1D**, and ATG8F, **Fig. 4C and F**), indicating loss of ATG4 activity.

Third, the lack of autophagic activity in the *atg4a/b* KO was further corroborated by GFP-cleavage assay (**Fig. 4D and Fig. 4E**).

Fourth, *atg4a/b* KO plants exhibit the typical autophagy-deficient phenotypes, e.g., early senescence, susceptibility to nutrient depletion, and stunted growth at later developmental stages (**Fig. 3, Fig. 4I-K**).

Fifth, the autophagy-deficient phenotype and lack of ATG8 lipidation in this *atg4a/b* line was independently shown by another group (<https://doi.org/10.1111/j.1365-3113X.2010.04166.x>).

Finally, we corroborated our findings using another ATG4 loss-of-function mutant, *atg4a-1/b-1*, established in a Wassilewskija-0 ecotype and utilized in the original study on the role of ATG8 delipidation in Arabidopsis by Yoshimoto and colleagues (doi: [10.1105/tpc.104.025395](https://doi.org/10.1105/tpc.104.025395)).

Comment 1.7: The quantification of autophagosomes and the western blot analysis for the level of free-EosFP were essential in Figure 2C.

Reply 1.7: We agree with the reviewer and performed the quantifications. Please refer to the updated **Figure 2** and **Fig. S3C and D**.

Comment 1.8: Some quantifications are required for the immune-blot, such as Figure 2D-E, Figure 4D-E, Figure S2D-E, and Figure S4C-D.

Reply 1.8: We agree with the reviewer and performed densitometry assays on all Western blots presented in the study.

Comment 1.9: Figure 5, the failure of the truncated ATG8 version to rescue the early senescence phenotype in the *cratg4* mutant may be attributed to deficiencies in autophagy rather than abnormal ATG8 delipidation. Firstly, it is necessary to verify whether autophagy flux is restored in *cratg4* by complementing it with mCherry-tagged full-length CrATG8.

Reply 1.9: Please refer to **Fig. 6**, the mCherry-CrATG8 expressed in the *cratg4* is shown on panels A and B.

Comment 1.10: If delipidation is not necessary, we should not find the autophagy-related phenotype in the *atg4a/atg4b* mutant, as ATG8H and ATG8I should still be functional. Here, the author further suggested that “...the sum amount of ATG8H and ATG8I is much lower than the sum amount of all isoforms ATG8A-ATG8I available for lipidation in the WT background and might not be sufficient for normal autophagosome biogenesis”, or “ ATG8H and ATG8I isoforms play specific roles in plant autophagy that do not include participation in autophagosome biogenesis”. However, these hypotheses are not sustained and require further confirmation.

Reply 1.10: It is indeed a very important and interesting point. In this study, we empirically demonstrated that stable overexpression of ATG8I only partially restores autophagic activity in the *atg4a/b* and yields autophagic bodies of a smaller size than observed when overexpressing the artificially primed ATG8E isoform. These results point out that amount of ATG8I is not the only reason for the autophagy-deficient phenotype of *atg4a/b*. Furthermore, we demonstrated that overexpression of ATG8I in WT triggers formation of smaller autophagic bodies under nitrogen depleted conditions, indicating specialization of ATG8I. Further elucidation of functionalization of ATG8 orthologs is indeed a very interesting topic, which we aim to pursue in our future projects.

Reviewer #2

Comment 2.1: Line 82-84: The manuscript states that ATG8E Δ C cannot be delipidated in the absence of ATG4. It is too early to state this without supporting data. It would be beneficial for the authors to modify this and let the readers to get the point after reading the related data (e.g. Fig. S4) presented later on.

Reply 2.1: Thank you for this suggestion, it was indeed meant as a theoretically predicted phenotype. We modified the text accordingly, please refer to Lines 95-97.

Comment 2.2: Majority of Western blot images presented are without size markers.

Reply 2.2: Thank you for bringing up this issue, we have added the MW markers to all Western blots.

Comment 2.3: A positive control demonstrating the effectiveness of NEM or IAM should be included In Figure 2C to validate the experimental results.

Reply 2.3: We added a Western blot showing *in vivo* accumulation of uncleaved EosFP-ATG8-CFP in plants treated with NEM and IAM. Please refer to the new Fig. S3D.

Comment 2.4: The statement in line 161-162 ‘... normally-functioning autophagosome formation and delivery to the vacuole under these conditions.’ may be bit too strong. Figure 2B indicates that there is a decrease in ATG8e Δ C delivery to the vacuole in *atg4/b* mutants, at least in shoots?

Reply 2.4: We appreciate the reviewer highlighting this aspect. Initially, we did not perceive the noted difference as biologically significant due to its minor nature, its presence solely in one of the two examined plant organs and it not recurring in subsequent experiments conducted under similar conditions (please see the updated Fig. S3F, blue boxes for AZD + ConA treatment). We noticed a potential issue with the statistical methods, causing exaggeration of statistical significance due to the high number of technical replicates and yielding misleading results. In the revised version we opted to average the data by biological replicates. This adjustment helped mitigate the influence of inherent variation in puncta density within the same plant organ on our analysis. Both the number of biological and technical replicates used for the analysis are specified in the figure legends.

Comment 2.5: I suggest changing the title to ‘ATG8 delipidation is dispensable for autophagy in Arabidopsis’, as there were no plants other than Arabidopsis tested.

Reply 2.5: We agree with the reviewer and edited the title of our study to better reflect main results and conclusions.

Comment 2.6: Numerous references in the manuscript are incorrectly formatted and require correction.

Reply 2.6: Thank you for bringing this up, we corrected the format of the references

REVIEWER COMMENTS

Reviewer #1 (Remarks to the Author):

The authors have addressed most of my previous concerns, but one major issue remains. Please refer to my earlier comment 3, the conclusions related to the requirement of ATG8 delipidation in Arabidopsis were mostly based on the observation of various over-expressing lines. However, these results may be used as clues, but ideally not as evidence. Since the authors already have the atg8 nonuple knockout lines available (I acknowledge it is tough to create this KO mutant), it would be nice to do further complementation assays to prove the authors' conclusion genetically. Could the ATG8 Δ C or ATG8i complement (at least partially) the stress sensitivity phenotype of atg8s? Only showing the localisation of ATG8 Δ C in the atg8s mutant background is not enough. This study should be part of the main figure and is the key experiment to nail it down.

Reviewer #2 (Remarks to the Author):

the authors addressed all my concerns. Personally I think the manuscript is acceptable now.

We would like to once again express our sincere gratitude to the reviewers for their valuable and constructive feedback on our manuscript. In response to Reviewer #1's request, we have performed the phenotyping experiments and, in addition, included two supplementary assays demonstrating the restoration of autophagic activity through the expression of the artificially truncated ATG8 in the ATG8 loss-of-function mutant. These new data, along with the previously presented results on the ATG8 loss-of-function mutant (originally detailed in our rebuttal letter), can now be found in Fig. 3, Fig. S6, and in Lines 205-213 of the revised manuscript (version with highlighted edits).